# CLUTR: Curriculum Learning via Unsupervised Task Representation Learning

## Abstract

Reinforcement Learning (RL) algorithms are often known for sample inefficiency and difficult generalization. Recently, Unsupervised Environment Design (UED) emerged as a new paradigm for zero-shot generalization by simultaneously learning a task distribution and agent policies on the sampled tasks. This is a non-stationary process where the task distribution evolves along with agent policies; creating an instability over time. While past works demonstrated the potential of such approaches, sampling effectively from the task space remains an open challenge, bottlenecking these approaches. To this end, we introduce CLUTR: a novel curriculum learning algorithm that decouples task representation and curriculum learning into a two-stage optimization. It first trains a recurrent variational autoencoder on randomly generated tasks to learn a latent task manifold. Next, a teacher agent creates a curriculum by maximizing a minimax REGRET-based objective on a set of latent tasks sampled from this manifold. By keeping the task manifold fixed, we show that CLUTR successfully overcomes the non-stationarity problem and improves stability. Our experimental results show CLUTR outperforms PAIRED, a principled and popular UED method, in terms of generalization and sample efficiency in the challenging CarRacing and navigation environments: showing an 18x improvement on the F1 CarRacing benchmark. CLUTR also performs comparably to the non-UED state-of-the-art for CarRacing, outperforming it in nine of the 20 tracks. CLUTR also achieves a 33% higher solved rate than PAIRED on a set of 18 out-of-distribution navigation tasks.

## 1 Introduction

Deep Reinforcement Learning (RL) has shown exciting progress in the past decade solving many challenging domains including Atari (Mnih et al. (2015)), Dota (Berner et al. (2019)), Go (Silver et al. (2016)). However, deep RL is sample-inefficient. Moreover, out-of-box deep RL agents are often brittle: performing poorly on tasks that they have not encountered during training, or often failing to solve them altogether even with the slightest change ( Cobbe et al. (2019), Azad et al. (2022), Zhang et al. (2018)). Curriculum Learning (CL) algorithms showed promise to improve (Portelas et al. (2020), Narvekar et al. (2020)) RL sample efficiency by employing a teacher algorithm that attempts to train the agents on tasks falling at the boundary of their capabilities, i.e., tasks that are slightly harder than the agents can currently solve. Recently, a class of unsupervised CL algorithms, called Unsupervised Environment Design (UED) [Dennis et al. (2020),Jiang et al. (2021a)], has shown impressive generalization capabilities which require no training tasks as input. UEDs automatically generate tasks by sampling from the free parameters of the environment (e.g., the start, goal, and obstacle locations for a navigation task) and attempt to improve sample efficiency and generalization by adapting a diverse task distribution at the agent's frontier of capabilities.

Protagonist Antagonist Induced Regret Environment Design (PAIRED) (Dennis et al. (2020)) is one of the most principled UED algorithms. The PAIRED teacher is itself an RL agent with actions denoting different task parameters. PAIRED aims at generating tasks that maximize the agent's regret, defined as the performance gap between an optimal policy and the student agent. Theoretically, upon convergence, the agent learns to minimize the regret, i.e., will solve every solvable task. Such a robustness guarantee makes regret-based teachers well suited for training robust agents.

Despite the strong robustness guarantee, PAIRED is still sample inefficient in practice. Primarily because training a regret-based teacher is hard (Parker-Holder et al. (2022)). First, the teacher receives a sparse reward only after specifying the full parameterization of a task; leading to a long-horizon credit assignment problem. Additionally, the teacher agent faces a combinatorial explosion problem if the parameter space is permutation invariant—e.g., for a navigation task, a set of obstacles corresponds to factorially different permutations of the parameters[1]. More importantly, to generate tasks at the frontier of agents' capabilities, the teacher needs to simultaneously learn a task manifold and navigate it to induce a curriculum. The teacher learns this task manifold implicitly based on regret. However, as the student is continuously co-learning with the teacher, the task manifold is also evolving over time. Hence, the teacher needs to simultaneously learn the evolving task manifold, as well as how to navigate it effectively—which is a difficult learning problem.

To address the above-mentioned challenges, we present Curriculum Learning via Unsupervised Task Representation Learning (CLUTR). At the core of CLUTR, lies a hierarchical graphical model that decouples task representation learning from curriculum learning. We develop a variational approximation to this problem and train a Recurrent Variational AutoEncoder (VAE) to learn a latent task manifold. Unlike PAIRED, which builds the tasks from scratch one parameter at a time, the CLUTR teacher generates tasks in a single timestep by sampling points from the latent task manifold and uses the generative model to translate them into complete tasks. The CLUTR teacher learns the curriculum by navigating the pretrained and fixed task manifold via maximizing regret. By utilizing a pretrained latent task-manifold, the CLUTR teacher can train as contextual bandit – overcoming the long-horizon credit assignment problem – and create a curriculum much more efficiently – improving stability at no cost to its effectiveness. Finally, by carefully introducing bias to the training corpus (such as sorting each parameter vector), CLUTR solves the combinatorial explosion problem of parameter space without using any costly environment interaction.

Our experimental results show that CLUTR outperforms PAIRED, both in terms of generalization and sample efficiency, in the challenging pixel-based continuous CarRacing and partially observable discrete navigation tasks. In CarRacing, CLUTR achieves 18x higher zero-shot generalization returns than PAIRED, while being trained on 60% fewer environment interactions on the F1 benchmark, modeled on real-life F1 racing tracks. Furthermore, CLUTR performs comparably to the non-UED attention-based SOTA(Tang et al. (2020)), outperforming it in nine of the 20 test tracks while requiring fewer than 1% of its environment interactions. In navigation tasks, CLUTR achieves higher zero-shot generalization in 14 out of the 18 test tasks, achieving a 33% higher solved rate overall. Furthermore, we empirically validate our hypotheses to justify the algorithmic decisions choices behind CLUTR.

In summary, we make the following contributions: i) we introduce CLUTR, a novel UED algorithm by augmenting the PAIRED teacher with unsupervised task-representation learning that is derived from a hierarchical graphical model for curriculum learning, ii) CLUTR, by decoupling task representation learning from curriculum learning, solves the long-horizon credit assignment and the combinatorial explosion problems faced by PAIRED. iii) Our experimental results show CLUTR outperforms PAIRED, both in terms of generalization and sample efficiency, in two challenging sets of tasks: CarRacing and navigation.

## 2 RELATED WORK

**Unsupervised Curriculum Design:** Dennis et al. (2020) was the first to formalize UED and introduced the minimax regret-based UED teacher algorithm, PAIRED with a strong theoretical robustness guarantee. However, gradient-based multi-agent RL has no convergence guarantees and often fails to converge in practice(Mazumdar et al. (2019)). Pre-existing techniques like Domain Randomization (DR) (Jakobi (1997), Sadeghi & Levine (2016), Tobin et al. (2017)) and minimax adversarial curriculum learning( Morimoto & Doya (2005), Pinto et al. (2017)) also fall under the category of UEDs. DR teacher follows a uniform random strategy, while the minimax adversarial teachers follow the maximin criteria, i.e., generate tasks that minimize the returns of the agent.

---

[1]Consider a 13x13 grid for a navigation task, where the locations are numbered from 1 to 169. Also consider a wall made of four obstacles spanning the locations: {21, 22, 23, 24}. This wall can be represented using any permutation of this set, e.g., {22, 24, 23, 21}, {23, 21, 24, 22}, resulting in a combinatorial explosion.

POET(Wang et al. (2019)) and Enhanced POET(Wang et al. (2020)) also approached UED, before PAIRED, using an evolutionary approach using a co-evolving population of tasks and agents.

Recently, Jiang et al. (2021a) proposed Dual Curriculum Design (DCD): a novel class of UEDs that augments UED generation methods (e.g., DR and PAIRED) with replay capabilities. DCD involves two teachers: one that actively generates tasks with PAIRED or DR, while the other curates the curriculum to replay previously generated tasks with Prioritized Level Replay(PLR)(Jiang et al. (2021b)). Jiang et al. (2021a) shows that, even with random generation (i.e., DR), updating the students only on the replayed level (but not while they are first generated, i.e., no exploratory student gradient updates as PLR) and with a regret-based scoring function, PLR can also learn minimax-regret agents at Nash Equilibrium and call this variation Robust PLR. It also introduces REPAIRED, combining PAIRED with Robust PLR. Parker-Holder et al. (2022) introduces ACCEL, which improves on Robust PLR by allowing edit/mutation of the tasks with an evolutionary algorithm. While CLUTR and PAIRED-variants actively adapt task generation to the performance of agents, other algorithms such as PLR generates task from a fixed task distribution. Different from PAIRED-variants, which are susceptible to instability due to evolving task-manifold, CLUTR introduces a novel variational formulation with a VAE-style pretraining for task-manifold learning.

**Representation Learning:** Variational Auto Encoders (Kingma & Welling (2013), Rezende et al. (2014), Higgins et al. (2016)) have widely been used for their ability to capture high-level semantic information from low-level data and generative properties in a wide variety of complex domains such as computer vision (Razavi et al. (2019), Gulrajani et al. (2016), Zhang et al. (2021), Zhang et al. (2022)), natural language (Bowman et al. (2015), Jain et al. (2017)), speech (Chorowski et al. (2019)), and music (Jiang et al. (2020)). VAE has been used in RL as well for representing image observations (Kendall et al. (2019), Yarats et al. (2021)) and generating goals (Nair et al. (2018)). While CLUTR also utilizes similar VAEs, different from prior work, it combines them in a new curriculum learning algorithm to learn a latent task manifold. Florensa et al. (2018) proposed a curriculum learning algorithm with latent-space goal generation using a Generative Adversarial Network.

## 3 BACKGROUND

### 3.1 UNSUPERVISED ENVIRONMENT DESIGN (UED)

As introduced by Dennis et al. (2020) UED is the problem of inducing a curriculum by designing a distribution of concrete, fully-specified environments, from an underspecified environment with free parameters. The fully specified environments are represented using a Partially Observable Markov Decision Process (POMDP) represented by $(A, O, S, \mathcal{T}, \mathcal{I}, \mathcal{R}, \gamma)$, where $A$, $O$, and $S$ denote the action, observation, and state spaces, respectively. $\mathcal{I} \rightarrow O$ is the observation function, $\mathcal{R} : S \rightarrow \mathbb{R}$ is the reward function, $\mathcal{T} : S \times A \rightarrow \Delta(S)$ is the transition function and $\gamma$ is the discount factor. The underspecified environments are defined in terms of an Underspecified Partially Observable Markov Decision Process (UPOMDP) represented by the tuple $\mathcal{M} = (A, O, \Theta, S^{\mathcal{M}}, \mathcal{T}^{\mathcal{M}}, \mathcal{I}^{\mathcal{M}}, \mathcal{R}^{\mathcal{M}}, \gamma)$. $\Theta$ is a set representing the free parameters of the environment and is incorporated in the transition function as $\mathcal{T}^{\mathcal{M}} : S \times A \times \Theta \rightarrow \Delta(S)$. Assigning a value to $\vec{\theta}$ results in a regular POMDP, i.e., UPOMDP + $\vec{\theta}$ = POMDP. Traditionally (e.g., in Dennis et al. (2020) and Jiang et al. (2021a)) $\Theta$ is considered as a trajectory of environment parameters $\vec{\theta}$ or just $\theta$—which we call task in this paper. For example, $\theta$ can be a concrete navigation task represented by a sequence of obstacle locations. We denote a concrete environment generated with the parameter $\vec{\theta} \in \Theta$ as $\mathcal{M}_{\vec{\theta}}$ or simply $\mathcal{M}_\theta$. The value of a policy $\pi$ in $\mathcal{M}_\theta$ is defined as $V^\theta(\pi) = \mathbb{E}[\sum_{t=0}^{T} r_t \gamma^t]$, where $r_t$ is the discounted reward obtained by $\pi$ in $\mathcal{M}_\theta$.

### 3.2 PAIRED

PAIRED Dennis et al. (2020) solves UED with an adversarial game involving three players [2]: the agent $\pi_P$ and an antagonist $\pi_A$, are trained on tasks generated by the teacher $\tilde{\theta}$. PAIRED objective

---

[2]In the original PAIRED paper, the primary student agent was named *protagonist*. Throughout this paper we refer it simply as the *agent*.

is: $max_{\tilde{\theta},\pi_P}min_{\pi_A}U(\pi_P,\pi_A,\tilde{\theta}) = \mathbb{E}_{\theta \sim \tilde{\theta}}[\text{REGRET}^\theta(\pi_P,\pi_A)]$. Regret is defined by the difference of the discounted rewards obtained by the antagonist and the agent in the generated tasks, i.e., $\text{REGRET}^\theta(\pi_P,\pi_A) = V^\theta(\pi_A) - V^\theta(\pi_P)$. The PAIRED teacher agent is defined as $\Lambda : \Pi \rightarrow \Delta(\Theta^T)$, where $\Pi$ is a set of possible agent policies and $\Theta^T$ is the set of possible tasks. The teacher is trained with an RL algorithm with $U$ as the reward while, the protagonist and antagonist agents are trained using the usual discounted rewards from the environments. Dennis et al. (2020) also introduced the flexible regret objective, an alternate regret approximation that is less susceptible to local optima. It is defined by the difference between the average score of the agent and antagonist returns and the score of the policy that achieved the highest average return.

## 4 CURRICULUM LEARNING VIA UNSUPERVISED TASK REPRESENTATION LEARNING

In this section, we formally present CLUTR as a latent UED and discuss it in details.

### 4.1 FORMULATION OF CLUTR

At the core of CLUTR is the latent generative model representing the latent task manifold. Let's assume that $R$ is a random variable that denotes a measure of success over the agent and antagonist agent and $z$ be a latent random variable that generates environments/tasks, denoted by the random variable $E$. We use the graphical model shown in Figure-1 to formulate CLUTR. Both $E$ and $R$ are observed variables while $z$ is an unobserved latent variable. $R$ can cover a broad range of measures used in different UED methods including PAIRED and DR (Domain Randomization). In PAIRED, $R$ represents the REGRET.

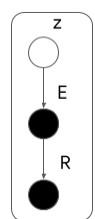

Figure 1: Hierarchical Graphical Model for CLUTR

We use a variational formulation of UED by using the above graphical model to derive the following ELBO for CLUTR, where $VAE(z,E)$ denotes the VAE objective:

$$ELBO \approx VAE(z,E) - \text{REGRET}(R,E) \tag{1}$$

We share the details of this derivation in Section B.1 of the Appendix. The above ELBO (Eq.1) defines the optimization objective for CLUTR, which can be seen as optimizing the VAE objective with a regret-based regularization term and vice versa. As previously discussed, it is difficult to train a UED teacher while jointly optimizing for both the curriculum and task representations. Hence we propose a two-level optimization for CLUTR. First, we pretrain a VAE to learn unsupervised task representations, and then in the curriculum learning phase, we optimize for regret to generate the curriculum while keeping the VAE fixed. In Section 5.4, we empirically show that this two-level optimization performs better than the joint optimization of Eq.1, i.e., finetuning the VAE decoder with the regret loss during the curriculum learning phase.

### 4.2 UNSUPERVISED LATENT TASK REPRESENTATION LEARNING

As discussed above, we use a Variational AutoEncoder(VAE) to model our generative latent task-manifold. Aligning with Dennis et al. (2020) and Jiang et al. (2021a), we represent task $\theta$, as a sequence of integers. For example, in a navigation task, these integers denote obstacle, agent, and goal locations. We use an LSTM-based Recurrent VAE (Bowman et al. (2015)) to learn task representations from integer sequences. We learn an embedding for each integer and use cross-entropy over the sequences to measure the reconstruction error. This design choice makes CLUTR applicable to task parameterization beyond integer sequences, e.g., to sentences or images. To train our VAEs, we generate random tasks by uniformly sampling from $\Theta^T$, the set of possible tasks. Thus, we do not require any interaction with the environment to learn the task manifold. Such unsupervised training of the task manifold is practically very useful as interactions with the environment/simulator are much more costly than sampling. Furthermore, we sort the input sequences, fully or partially, when they are permutation invariant, i.e., essentially represent a set. By sorting the training sequences, we thus avoid the combinatorial explosion faced by the PAIRED teacher.

| Algorithm | Task Representation Learning | Teacher Model | UED Method | Replay Method |
|---|---|---|---|---|
| DR | | | | - |
| PLR | | Random | DR | PLR |
| Robust PLR | - | | | Robust PLR |
| ACCEL | | | DR + Evolution | |
| PAIRED | Implicit via RL | Learned | PAIRED | - |
| REPAIRED | | | | Robust PLR |
| CLUTR | Explicit via Unsupervised Generative Model | | | - |

Table 1: A comparative characterization of contemporary UED methods

## 4.3 CLUTR

We define CLUTR following the objective given in Eq. 1. CLUTR uses the same curriculum objective as PAIRED, $\text{REGRET}(R, E) = \text{REGRET}^\theta(\pi_P, \pi_A)$ where, $\theta$ denotes a task, i.e., a concrete assignment to the free parameters of the environment $E$. Unlike PAIRED teacher, which generates $\theta$ directly, the CLUTR teacher policy is defined as $\Lambda : \Pi \to \Delta(\mathcal{Z})$, where $\Pi$ is a set of possible agent policies and $\mathcal{Z}$ is as the latent space. Thus, the CLUTR teacher is a latent environment designer, which samples random $z$ and $\theta$ is generated by the VAE decoder function $\mathcal{G} : \mathcal{Z} \to \Theta$. We present the outline of the CLUTR in Algorithm 1. CLUTR outline is very similar to PAIRED, differing only in the first two lines of the main loop to incorporate the latent space.

---
**Algorithm 1** CLUTR
---
Pretrain VAE with randomly sampled tasks from $\Theta$
Randomly initialize Agent $\pi^P$, Antagonist $\pi^A$, and Teacher $\tilde{\Lambda}$;
**while** *not converged* **do**
    Generate latent task vector: $z \sim \mathcal{Z}$ from the teacher
    Create POMDP $M_\theta$ where $\theta = \mathcal{G}(z)$ and $\mathcal{G}$ is the VAE decoder function
    Collect Agent trajectory $\tau^P$ in $M_\theta$. Compute: $U^\theta(\pi^P) = \sum_{i=0}^{T} r_t \gamma^t$
    Collect Antagonist trajectory $\tau^A$ in $M_\theta$. Compute: $U^\theta(\pi^A) = \sum_{i=0}^{T} r_t \gamma^t$
    Compute: $\text{REGRET}^\theta(\pi^P, \pi^A) = U^\theta(\pi^A) - U^\theta(\pi^P)$
    Train Protagonist policy $\pi^P$ with RL update and reward $R(\tau^P) = U^\theta(\pi^P)$
    Train Antagonist policy $\pi^A$ with RL update and reward $R(\tau^A) = U^\theta(\pi^A)$
    Train Teacher policy $\tilde{\Lambda}$ with RL update and reward $R(\tau^{\tilde{\Lambda}}) = \text{REGRET}$
**end while**
---

Now we discuss a couple of additional properties of CLUTR compared to PAIRED-variants.

1. CLUTR teacher samples from the latent space $\mathcal{Z}$ and thus generates a task in a single timestep. Note that this is not possible for the PAIRED/REPAIRED teacher, as they generate one task parameter at a time, conditioned on the state of the partially-generated task so far.

2. PAIRED-variants typically observe the state of the partially generated task to generate the next parameter. Hence depending on the state space, they require designing different teacher architectures for different environments. CLUTR teacher architecture, however, is agnostic of the problem domain. Hence the same architecture can be used across different problems.

## 4.4 COMPARISON OF CLUTR WITH CONTEMPORARY UED METHODS

As discussed in 2, contemporary UED methods can be characterized by their i) teacher type (random/fixed or, learned/adaptive) and ii) the use of replay. To clearly place CLUTR in the context of contemporary UEDs, we discuss another important aspect of curriculum learning algorithms: how the task manifold is learned. The random UED teachers do not learn a task manifold. Regret-based

teachers such as PAIRED and REPAIRED learn an implicit (e.g., the hidden state of the teacher LSTM) task-manifold but it is not used explicitly. It is trained via RL based on regret estimates of the tasks they generate. Hence, these task-manifolds depend on the quality of the estimates, which in turn depends on the overall health of the multi-agent RL training. Furthermore, they do not take into account the actual task structures. In contrast, CLUTR introduces an explicit task-manifold modeled with VAE that presents a local neighborhood structure capturing the similarity of the tasks. Hence, similar tasks (in terms of parameterization) would be placed nearby in the latent space. Intuitively this local neighborhood structure should facilitate the teacher to navigate the manifold. The above discussion illustrates that CLUTR along with PAIRED and REPAIRED form a category of UEDs that generates tasks based on a learned task-manifold, orthogonal to the random generation-based methods, while CLUTR being the only one utilizing an unsupervised generative task manifold. Table 1 summarizes the differences.

## 5    EXPERIMENTS

In this section, we first evaluate CLUTR's performance compared to the existing UEDs in Pixel-Based Car Racing with continuous control and dense rewards. As discussed in Section 4.4, our primary comparison is with PAIRED and REPAIRED—the only two existing UED methods that learn task-manifolds to generate tasks. For completeness, we also compare CLUTR with UEDs with random teachers. Furthermore, we compare with PAIRED on partially observable navigation tasks with discrete control and sparse rewards.

We then empirically evaluate the following hypotheses:
**H1**: CLUTR generates a more efficient curriculum. (Section 5.3)
**H2**: Learning task representations and curriculum simultaneously degrades the performance (5.4)
**H3**: Training VAE on sorted data solves the combinatorial explosion problem. (Section 5.5)

At last, we analytically compare the CLUTR and PAIRED curricula. Full details of the environments, network architectures of the teacher and student agents, the VAE, the training hyperparameters, and further analysis and evaluation are discussed in the Appendix.

### 5.1    CLUTR PERFORMANCE ON PIXEL-BASED CONTINUOUS CONTROL CARRACING ENVIRONMENT

The CarRacing environment (Jiang et al. (2021a), Brockman et al. (2016)) requires the agent to drive a full lap around a closed-loop racing track modeled with Bézier Curves (Mortenson (1999)) of up to 12 control points. CLUTR was trained with the Flexible Regret Objective for 2M timesteps—around which the agent's training return converges to its maximum. We also experimented with the standard regret objective and obtained better performance than PAIRED, which we discussed in Section D.2 of the appendix. We train the VAE on 1 Million randomly generated tracks for 1 Million gradient updates. Note that only one VAE was trained and used for all the experiments (10 independent runs). We evaluate the agents on the F1 benchmark (Jiang et al. (2021a)) that contains 20 test tracks modeled on real-life F1 racing tracks. These tracks are significantly out of distribution than any tracks that the UED teachers can generate with just 12 control points. Further details on the environment, network architectures, and VAE training can be found in Section C.1,C.2, and C.4 of the appendix, respectively.

Figure 2a shows the mean return obtained by the CLUTR, PAIRED, and REPAIRED on the 20 F1 test tracks. CLUTR outperforms PAIRED and REPAIRED by a huge margin: showing an 18x higher mean return than PAIRED and 1.6x than REPAIRED, outperforming both of them in all of the 20 test tracks. Note that CLUTR was trained only for 2M timesteps, while both PAIRED and REPAIRED were trained for 5M timesteps. Figure 2b tracks the agents' generalization capabilities by periodically evaluating them on four unseen tracks Vanilla, Singapore, Germany, and Italy. These tracks were selected aligning with Jiang et al. (2021a). Based on these selected tracks, CLUTR shows much better generalization and sample efficiency—achieving better performance and faster improvement. Further experiment results are shared in Section D of the appendix.

We also compare CLUTR to other existing UEDs with random task generation on the F1 benchmark for completeness. CLUTR outperforms Domain Randomization and PLR, falling short only to Robust PLR, which achieves an overall 1.13X higher returns. Nonetheless, CLUTR performs

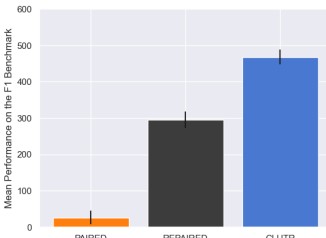 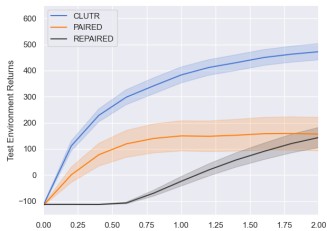 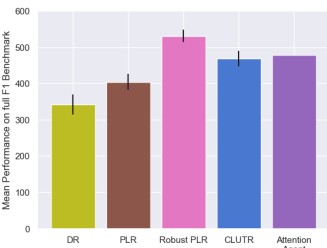

(a) Mean Performance on the F1 Benchmark. CLUTR achieves an 18x higher return than PAIRED and 1.6x than REPAIRED.

(b) Agent performance on selected test tracks during training. CLUTR shows significantly better generalization and sample efficiency.

(c) Mean Performance on the F1 benchmark, compared to other existing UEDs and the non-UED SOTA.

Figure 2: CLUTR performance on the CarRacing Tasks compared to PAIRED and REPAIRED. The results show mean and standard error of 10 independent runs. Further details can be found in Section D in Appendix

comparably to Robust PLR on seven of the 20 tracks and outperforms in one. Furthermore, CLUTR shows comparable performance to the non-UED attention-based state-of-the-art method for CarRacing (Tang et al. (2020)), despite not using a self-attention policy and training on significantly fewer environment timesteps ($< 1\%$). Moreover, CLUTR outperforms it on nine of the 20 tracks. Detailed results on individual tracks are presented in Table 4 of the Appendix.

## 5.2 CLUTR PERFORMANCE ON PARTIALLY OBSERVABLE NAVIGATION TASKS ON MINIGRID

We also compare CLUTR with PAIRED on the popular MiniGrid environment, originally introduced by Chevalier-Boisvert et al. (2018) and adopted by Dennis et al. (2020) for UEDs. In these navigation tasks, an agent explores a grid world to find the goal while avoiding obstacles and receives a sparse reward upon reaching the goal. To train CLUTR VAE, we generate 1 Million random grids, with the obstacle locations sorted, and the number of obstacles uniformly varying from zero to 50, aligning with Dennis et al. (2020). We used the standard regret objectives. Note that the results reported in the original PAIRED paper are obtained after 3 Billion

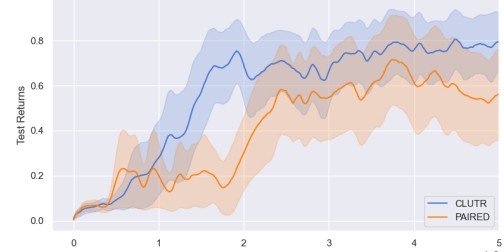

Figure 3: Agent solved rate on selected grids during training. CLUTR shows better sample efficiency and generalization than PAIRED. The results show an average of 5 independent runs.

timesteps of training, while we run both PAIRED and CLUTR for 500M timesteps (5 independent runs) due to the huge computational resource and time needed to run a training with 3 Billion timesteps. For the same computational constraints, we compare only with PAIRED in this environment. Figure 4 shows zero-shot generalization performance of CLUTR and PAIRED 18 unseen navigation tasks from Dennis et al. (2020) based on the percent of environments the agent solved, i.e., solved rate. CLUTR achieves superior generalization solving 64% of the unseen grids, while PAIRED achieves 43%, which is 33% lower compared to CLUTR. From figure 4 it can be seen CLUTR outperforms PAIRED achieving a higher mean solve rate on 14 out of the 18 test navigation tasks. Figure 3 shows solved rates on four selected grids (Sixteen Rooms, Sixteen Rooms with Fewer Doors, Labyrinth, and Large Corridor) during training. CLUTR shows better sample efficiency, as well as generalization than PAIRED.

## 5.3 EFFICIENCY OF THE CURRICULUM: CLUTR VS PAIRED

Figure 5 shows the mean regret on the teacher-generated tasks for both CarRacing and navigation tasks. CLUTR shows a lower regret than PAIRED, meaning the performance gap between the agent and the antagonist is lower in CLUTR. From a curriculum learning perspective, we want to train

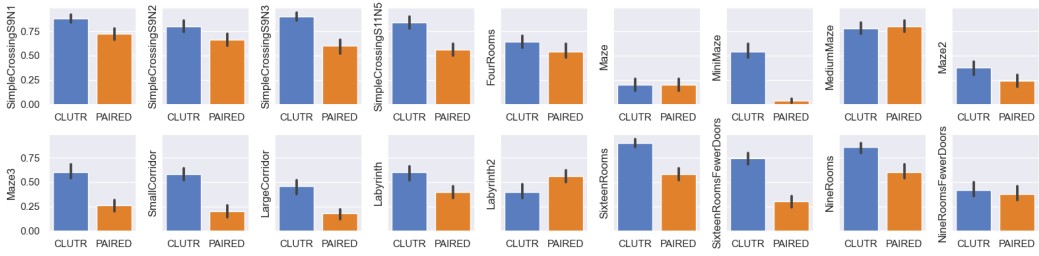

Figure 4: Zero-shot generalization of CLUTR and PAIRED, in terms of percent of the environments solved. CLUTR achieves a higher solved rate than PAIRED in 14 out of the 18 tasks. We evaluate the agents with 100 independent episodes on each task. Error bars denote the standard error.

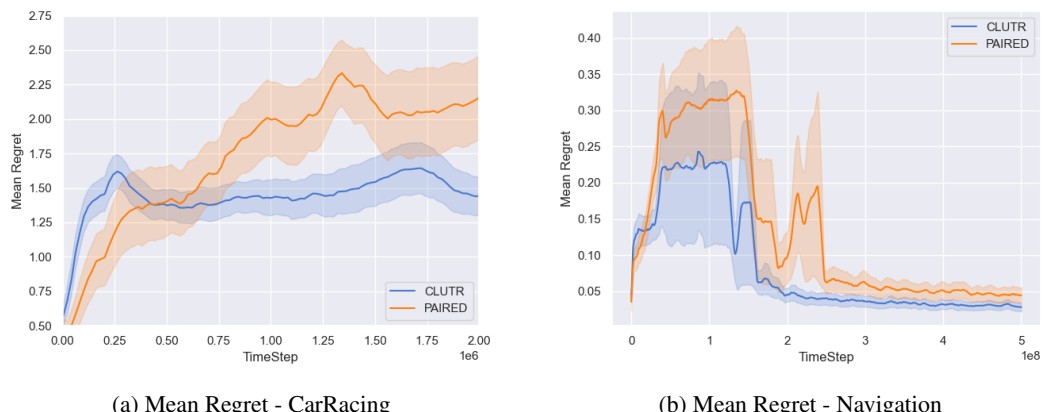

(a) Mean Regret - CarRacing

(b) Mean Regret - Navigation

Figure 5: Mean regret values during training. CLUTR shows a smaller regret value indicating a less performance gap between the agent and the antagonist. CLUTR also converges faster.

the agent on tasks that are slightly harder than it can already solve or, those tasks that it can solve already but can obtain better returns. In practice, both the agent and the antagonist are trained in the same training context e.g., the same hyper-parameters, model architecture, and tasks, differing only by their random initial weights. Hence, a lower regret means that the teacher is generating tasks that are either slightly harder than the tasks the agent can solve now (because the other agent is solving them) or, tasks in which the antagonist is performing slightly better. Hence, the tasks are more likely at the agent's frontier of capability. The curves also show that CLUTR and PAIRED show similar convergence patterns, while CLUTR converges sooner to a better local optimum. These observations, in addition to the empirical performance, indicate that CLUTR is generating a more efficient curriculum than PAIRED.

## 5.4 Learning task manifold and curriculum: Joint vs Two-staged Optimization

etuned We hypothesize that learning the task representations and the curriculum simultaneously results in a difficult training problem due to the non-stationarity of the task manifold. To test this, instead of keeping the task representations fixed, we continue finetuning our decoder on the regret loss during the teacher-student curriculum learning phase. This experiment shows a 58% performance drop in the F1 benchmark, labeled 'Finetuned VAE' in Figure 6. This empirically validates our hypothesis that pretraining a latent task space and then learning to navigate it to induce curriculum indeed is easier and can lend to better UED.

## 5.5 Impact of sorting VAE data on solving Combinatorial Explosion

We hypothesized that training a VAE on sorted sequences can solve the combinatorial explosion problem. To test this, we run CLUTR with an alternate VAE trained 5X longer

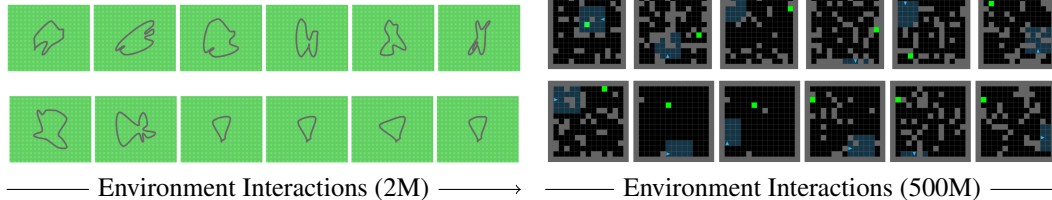

Figure 7: Example tracks(left) and grids(right) generated by CLUTR(top) and PAIRED(bottom) uniformly sampled at different stages of training. The training progresses from left to right.

on a non-sorted and 10X bigger version of the original dataset. This experiment shows a 59% performance drop on the F1 benchmark, labeled 'Shuffled VAE' in Figure 6, empirically validating our hypothesis. Further details are discussed in Section D.3 of the Appendix.

## 5.6 CURRICULUM COMPLEXITY

In this section, we compare the curriculum generated by CLUTR and PAIRED, with snapshots of tasks generated by these methods during different stages of the training (Figure 7). We illustrate one common mode of failure/ineffectiveness shown by PAIRED: The curriculum starts with arbitrarily complex tasks, which none of the agents can solve at the initial stage of training. After a while, PAIRED starts generating rudimentary degenerate tasks. If enough training budget is given, PAIRED eventually gets out of the degenerative local minima, and the curriculum complexity starts to emerge. On the other hand, CLUTR does not show such degeneration and generates seemingly interesting tasks throughout. The examples shown Figure 7 illustrates this.

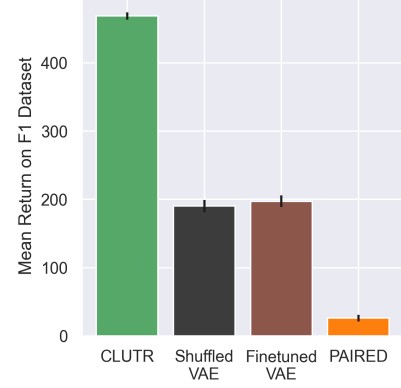

Figure 6: Impact of i) Training VAE on non-sorted data (Shuffled VAE) and ii) Finetuning the task-manifold with regret (Finetuned VAE) on F1 benchmark.

## 6 CONCLUSION AND FUTURE WORK

In this work, we propose CLUTR, an unsupervised environment design method via unsupervised task representation learning. CLUTR augments PAIRED with a latent task space, decoupling task representation learning from curriculum learning. CLUTR poses several advantages over PAIRED-variants, including solving the long-horizon credit assignment and the combinatorial explosion of the parameter space. Our experiments show CLUTR outperforms PAIRED-variants in terms of sample efficiency and generalization.

Even though CLUTR and other regret-based UEDs empirically show good generalization on human-curated complex transfer tasks, they rarely can generate human-level task structures during training. An interesting direction would be to enable UED algorithms to generate realistic tasks. Another important direction would be to reduce the gap between the theoretical and practical aspects of regret-based multi-agent UED algorithms, which are subject to the quality of regret estimates and multi-agent RL training. At last, random generator algorithms like Robust PLR or even, DR have been shown to perform better than learned generator approaches like CLUTR or PAIRED. An interesting direction would be to investigate the conditions/environments under which a random generator performs better than an adaptive generator and vice versa. At last, we are excited about latent-space curriculum design and hope our work will encourage further research in this domain.

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

## ETHIC STATEMENT

Unsupervised Environment Design can be applied to many real-world applications and shares many similar ethical concerns and considerations with other Artificially Intelligent(AI) systems. For example, AI systems can cause more unemployment or be used for reasons/applications that have a negative societal impact, for which responsible usage of such AI systems must be promoted and established. During our research, all the experiments were done in simulation and no human or living subjects were used.

## REPRODUCIBILITY

Our code, saved checkpoints, and training data are available at `https://github.com/clutr/clutr`

## A    APPENDIX

## B    ADDITIONAL DETAILS OF CLUTR

### B.1    CLUTR OBJECTIVE DERIVATION

We use a hierarchical graphical model to formulate the latent environment design problem. Let's assume that $R$ is a random variable that denotes a measure of success defined using the protagonist and antagonist agents and $z$ be a latent random variable. We use the graphical model in Figure-8 where $z$ generates an environment $E$ and $R$ is the success defined over $E$. Both $E$ and $R$ are observed variables while $z$ is an unobserved variable. $R$ covers a broad range of measures used in different UED methods including PAIRED and DR (Domain Randomization). In PAIRED, $R$ represents the REGRET as the difference of returns between the antagonist and protagonist agents and it depends on the environments that the agents are evaluated on.

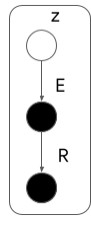

Figure 8: Hierarchical Graphical Model for CLUTR

We use a variational formulation of UED by using the above graphical model. We first define the variational objective as the KL-divergence between an approximate posterior distribution and true posterior distribution over latent variable $z$,

$$D_{KL}(q(z)|p(z|R,E)) = E_{z \sim q(z)}[log q(z)] - E_{z \sim q(z)}[log p(z|R,E)]$$
$$= E_{z \sim q(z)}[log q(z)] - E_{z \sim q(z)}[log p(R,E,z)] + log p(R,E)$$

where both $R$ and $E$ are given.

Next, we write the ELBO,

$$ELBO = E_{z \sim q(z)}[log q(z)] - E_{z \sim q(z)}[log p(R,E,z)]$$
$$= E_{z \sim q(z)}[log q(z)] - E_{z \sim q(z)}[log p(R|E)p(E|z)p(z)]$$
$$= E_{z \sim q(z)}[log q(z)] - E_{z \sim q(z)}[log p(z)] - E_{z \sim q(z)}[log p(E|z)] - E_{z \sim q(z)}[log p(R|E)]$$
$$= E_{z \sim q(z)}[log \frac{q(z)}{p(z)}] - E_{z \sim q(z)}[log p(E|z)] - log p(R|E)$$
$$= D_{KL}(q(z)|p(z)) - E_{z \sim q(z)}[log p(E|z)] - log p(R|E)$$
$$= VAE(z,E) - log p(R|E)$$

We can also induce an objective that includes minimax REGRET. Let $R$ be distributed according to an exponential distribution, $p(R|E) \propto exp(\text{REGRET}(\pi_P, \pi_A|E))$,

we derive,

$$ELBO \approx VAE(z,E) - \text{REGRET}(R,E)$$

where the normalizing factor is ignored.

### B.2    ROBUSTNESS GUARANTEES

CLUTR essentially proposes including a pretrained latent space within the teacher/generator. From the teacher's perspective, the difference is while the PAIRED teacher starts from randomly initialized weights, CLUTR starts from the pretrained weights. Thus, CLUTR does not impose new assumptions on possible teacher policies. Furthermore, CLUTR does not change any other specifics of the underlying PAIRED algorithm. Hence, CLUTR holds the same theoretical robustness guarantees provided by PAIRED.

In practice, both CLUTR and PAIRED deviate from these theoretical guarantees. For example, both algorithms approximate the regret value, which is the case for other regret-based UEDs such

as Robust PLR and REPAIRED (Jiang et al. (2021a)). Also, the robustness guarantee depends on reaching the Nash equilibrium of the multiagent adversarial game. However, gradient-based multi-agent RL has no convergence guarantees and often fails to converge in practice(Mazumdar et al. (2019)). We also note that, by introducing the latent space, CLUTR VAE might not have access to the full task space due to practical limitations on training, e.g., the training dataset not having all possible tasks. However, when the decoder is allowed to be finetuned, CLUTR will have access to the full task space, similar to PAIRED. Our empirical results (discussed in Section 5.4) suggest that keeping the pretrained decoder fixed performs better than finetuning it, so we kept it fixed for our main experiments. We also want to mention that we used the flexible regret objective for CarRacing in Section 5.1. When the flexible objective is used, CLUTR (and PAIRED) might not hold the robustness guarantee as it changes the dynamics of the underlying game between the teacher and the agents. However, we also experimented with the standard regret objective and obtained better performance than PAIRED as discussed in Section D.2.

## C  TRAINING DETAILS

### C.1  ENVIRONMENT DETAILS

**Car Racing**: The CarRacing environment was originally proposed by OpenAI Gym Brockman et al. (2016), and later has been reparameterized by Jiang et al. (2021a) with Bézier Curves( Mortenson (1999)) for UED algorithms. This environment requires the agents to drive a full lap around a closed-loop track. The track is defined by a Bézier Curve modeled with a sequence of upto 12 arbitrary control points, each spaced within a fixed radius $B/2$ of the center of the $B \times B$ field. This sequence of control points can uniquely identify a track, subject to a set of predefined curvature constraints Jiang et al. (2021a). The control points are encoded in a $10 \times 10$ grid—a discrete down-sampled version of the racing track field. Each control point hence is a integer denoting a cell of the grid and the cell coordinates are upscaled to match the original scale of the field afterwards. This ensures no two control points are too close together, preventing areas of excessive track overlapping. The track consists of a sequence of $L$ polygons and the agent receives a reward of $1000/L$ upon visiting each unvisited polygon and a penalty of $-0.1$ at each time step to incentivize completing the tracks faster. Episodes terminate if the agent drives too far off-track but is not given any additional penalty. The agent controls a 3 dimensional continuous action space corresponding to the car's steer: torque $\in [-1.0, 1.0]$, gas: acceleration $\in [0, 0, 1.0]$, and brake: deceleration $\in [0.0, 1.0]$. Each action is repeated 8 times. The agent receive a $96 \times 96 \times 3$ RGB pixel observation. The top $84 \times 96$ portion of the frame contains a clipped, egocentric, bird's eye view of the horizontally centered car. The bottom $12 \times 96$ segment simulates a dashboard visualizing the agent's latest action and return. Snapshots of the test track in the F1 benchmark are shown in Figure 9.

**Minigrid**: The environment is partially observable and based on Chevalier-Boisvert et al. (2018) and adopted for UED by Dennis et al. (2020). Each navigation task is represented with a sequence of integers denoting the locations of the obstacles, the goal, and the starting position of the agent: on a $15 \times 15$ grid similar to Dennis et al. (2020). The grids are surrounded by walls on the sides, making it essentially a $13 \times 13$ grid. Dennis et al. (2020) parameterizes the locations using integers. Each task is a sequence of 52 integers, while the first 50 numbers denote the location of obstacles followed by the goal and the agent's initial location. The sequences may contain duplicates to allow the generation of navigation tasks with fewer than 50 obstacles. Snapshots of the test grids used in our paper are shown in Figure 10.

### C.2  NETWORK ARCHITECTURES

All the student and teacher agents are trained with PPO Schulman et al. (2017).

**Student Architecture**

For CarRacing, we use the same student architecture as Jiang et al. (2021a). The architecture consists an image embedding module composed of 2D Convolutions with square kernels of sizes 2,2,2,2,3,3, stride lengths 2,2,2,2,1,1 and channel outputs of 8, 16, 64, 128, 256 stacked together. The image embedding is of size 256 and is passed through a Fully Connected (FC) layer of 100 hidden units and then passed through ReLU activations. This embedding is then passed through two FC with 100

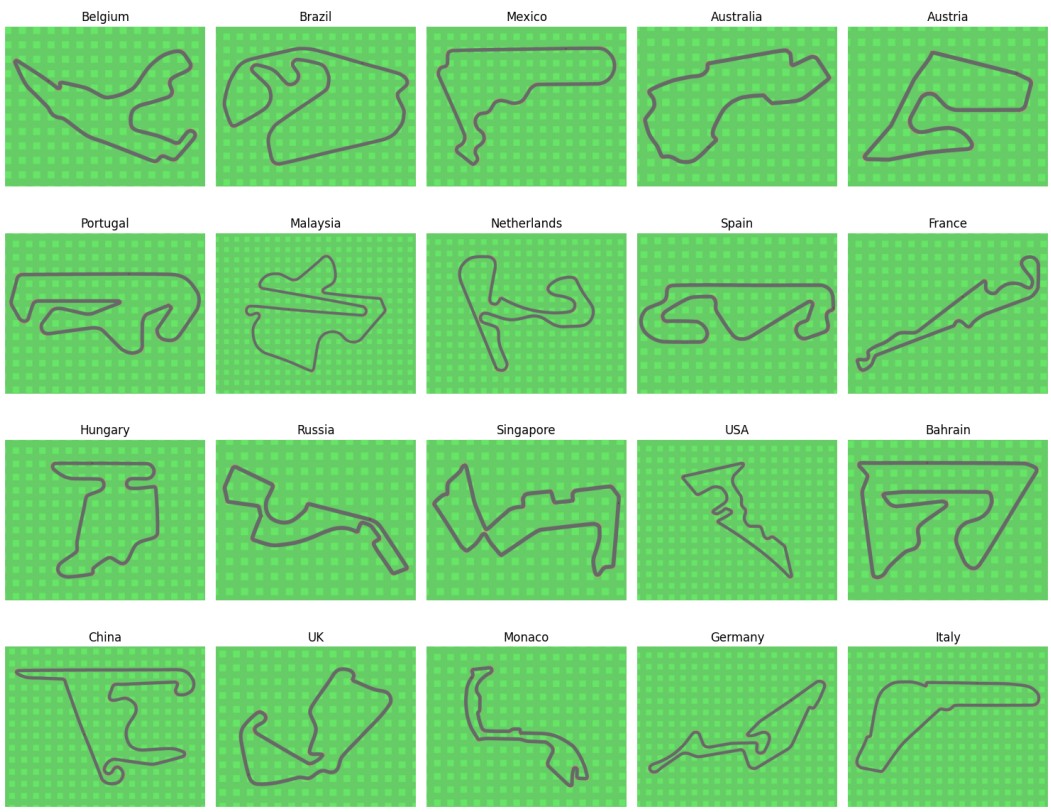

Figure 9: Snapshots of the test tracks in F1 benchmark

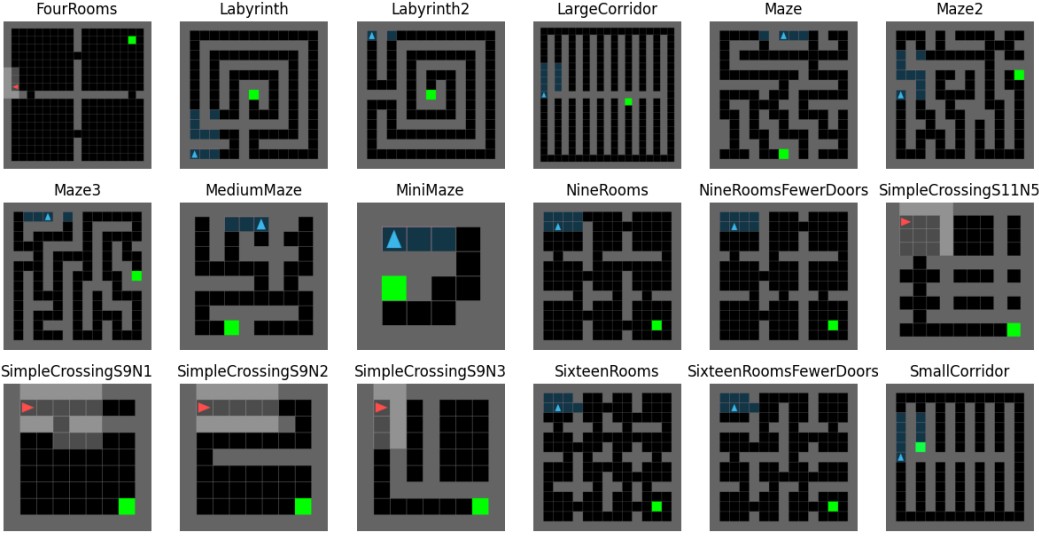

Figure 10: Snapshots of the test grids for MiniGrid

hidden neurons, and then a softplus layer, and finally added to 1 for the beta distribution used for the continuous action space. Further details can be found in Jiang et al. (2021a).

For navigation tasks, we use the same student architecture as Dennis et al. (2020). The observation is a tuple with a $5 \times 5 \times 3$ grid observation and a direction integer in $[0 - 3]$. The grid view is fed to a convolutional layer with kernels of size 3 with 16 filters and the direction integer is passed through a FC with 5 units. This is followed by an LSTM of size 256, and then to two FC layers with 32 units, which connect to the policy outputs. The value network uses the same architecture.

**Teacher Architecture**

For CarRacing, CLUTR teacher takes a random noise and generates a continuous vector, i.e., the latent task vector. We pass the random noise through a feed-forward network with one hidden layer of 8 neurons as the teacher. The output of this layer is fed through two separate fully-connected layers, each with a hidden size of 100 and an output dimension equal to the latent space dimension, followed by soft plus activations. We then add 1 to each component of these two output vectors, which serve as the $\alpha$ and $\beta$ parameters respectively for the Beta distributions used to sample each latent dimension. In our experiments, we used a 64-dimensional latent task space. For Minigrid experiments, we use a network architecture similar to Dennis et al. (2020) but take only the random noise as input. The adversary network generates discrete actions, but we map them to real numbers to feed into the VAE decoder.

**VAE architecture**

We use the architecture proposed in Bowman et al. (2015). We use a word-embedding layer of size 300 with random initialization. The encoder comprises a conditional 'Highway' network followed by an LSTM. The Highway network is a two-staged network stacked on top of each other. Each stage computes $\sigma(x) \odot f(G(x)) + (1 - \sigma(x)) \odot Q(x)$, where $x$ is the inputs to each of the highway network stages, G and Q is affine transformation, $\sigma(x)$ is a sigmoid non-linearization, and $\odot$ is element-wise multiplication. $G$ and $Q$ are feed-forward networks with a single hidden layer with equal input and output dimensions of 300, equal to the word-embedding output dimension. We use ReLU activation as $f$. The highway network is followed by a bidirectional LSTM with a single layer of 600 units. The LSTM outputs are passed through linear layer of dimension 64 to get the VAE mean and log variance. The mean vectors are passed through a hyperbolic tangent activation and for the navigation tasks linearly scaled in $[-4, 4]$. The decoder takes in latent vectors of dimension 64 and passes through a bidirectional LSTM with two hidden layers of size 800 and follows it by a linear layer with size equaling the parameter vector dimension.

### C.3 HYPERPARAMETERS

All our agents are trained with PPO (Schulman et al. (2017)). We did not perform any hyperparameter search for our experiments. The CarRacing experiments used the same parameters used in Jiang et al. (2021a), and the Minigrid experiments used the parameters from Dennis et al. (2020). VAE was trained on the parameters from Bowman et al. (2015). The detailed parameters are listed in Table 2 and Table 3.

| Parameter | Value |
|---|---:|
| Batch Size | 32 |
| Number of Training Steps | 1000000 |
| Reconstruction Weight | 79 |
| Latent Variable Size | 64 |
| Word Embedding size | 300 |
| Maximum Sequence Length | 52 |
| Encoder Activation | Hyperbolic Tangent |
| Learning Rate | 0.00005 |
| Dropout | 0.3 |

Table 2: Hyperparameters for training the Task VAE

| Parameter | CarRacing | MiniGrid |
|---|---|---|
| $\gamma$ | 0.99 | 0.995 |
| $\lambda_{GAE}$ | 0.9 | 0.95 |
| PPO rollout length | 125 | 256 |
| PPO epochs | 8 | 5 |
| PPO minibatches per epoch | 4 | 1 |
| PPO clip range | 0.2 | 0.2 |
| PPO number of workers | 16 | 32 |
| Adam learning rate | 3e-4 | 1e-4 |
| Adam $\epsilon$ | 1e-5 | 1e-5 |
| PPO max gradient norm | 0.5 | 0.5 |
| PPO value clipping | no | yes |
| Return normalization | yes | no |
| Value loss coefficient | 0.5 | 0.5 |
| Student entropy coefficient | 0 | 0 |
| Action Repeat | 8 | - |

Table 3: Hyperparameters for PAIRED and CLUTR PPO training.

### C.4 VAE TRAINING DATA

For CarRacing, we follow the same parameterization as Jiang et al. (2021a): each track is defined with a sequence of up to 12 integers denoting control points of a Bézier Curve. . Each control point is represented with an integer. We generate 1M random sorted integer sequences of fixed length 12 with duplicates—which enables generating tracks defined with less than 12 control points. For navigation tasks we use the parameterization of Dennis et al. (2020), generating upto 50 obstacles for each task for a $15 \times 15$ grid, surrounded by walls, effectively an active area of $13 \times 13$. Hence, each location is numbered in 1 to 169. Every number except the last two of the sequence represent obstacle locations, and the last two for the goal and agent location, respectively. The parameter vector is thus partially permutation invariant. We uniformly generate 1M sequences of variable length between 2 and 52 (inclusive). The obstacle locations are sorted.

## D DETAILED RESULTS ON CARRACING

### D.1 DETAILED COMPARISON ON FULL F1 DATASET

We used the flexible regret approximation for the results presented in the main paper. The flexible regret objective is a more robust variant of the standard regret estimation (both introduced in Dennis et al. (2020)). It is defined by the difference between the average score of the agent and antagonist returns and the score of the policy that achieved the highest average return. Thus, the flexible objective blurs the distinction between the agent and the antagonist. Hence we designate the agent achieving the higher average training return during the last 10 steps as the agent.

Figure 11 compares how different UEDs perform during training by periodically evaluating them on Four Selected Tracks: Vanilla, Singapore, Germany, and Italy. These tracks were selected aligning with Jiang et al. (2021a). Based on these selected tracks, CLUTR performance plateaus around 2.5M timesteps. Robust PLR starts slowly but surpasses all the other methods after 5M timesteps.

Table 4 shows the comparison between the final agents trained with CLUTR and other UED algorithms. It is to be noted that, all the UED methods except CLUTR was trained for 5M timesteps where CLUTR was run for 2M timesteps. CLUTR outperforms PAIRED by a big margin with 18x bigger mean return on the entire F1 Dataset. CLUTR also outperforms Domain Randomization, PLR, and REPAIRED and only falls short to Robust PLR. Nonetheless, CLUTR shows competitive results compared to Robust PLR, showing comparable results in seven out of the 20 test tracks and outperforming in the Netherlands track. CLUTR also outperforms the non-UED SOTA on the full F1 dataset. CLUTR outperforms the Attention Agent on 9 out of the 20 tracks and shows comparable performance in one.

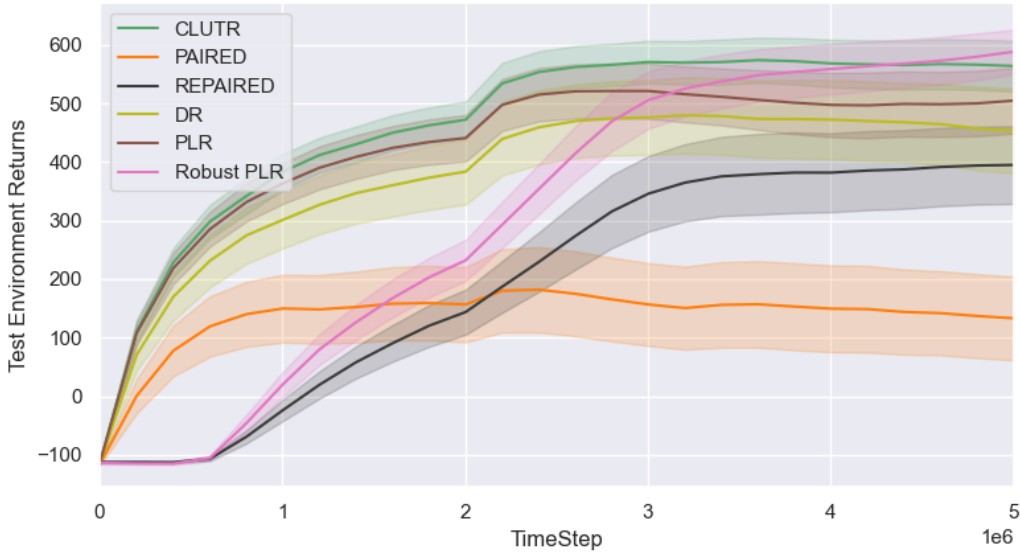

Figure 11: Comparison of mean agent returns on Four Selected Tracks: Vanilla, Singapore, Germany, and Italy. Based on these selected tracks, CLUTR improves a bit after 2M timesteps later the performance plateaus. Robust PLR starts slowly but surpasses all the other methods after 5M timesteps.

## D.2 CLUTR WITH STANDARD REGRET LOSS

We also train CLUTR with the standard regret loss for 5M timesteps. Figure 12 compares the impact of standard/flexible regret loss on the regret and agent returns during training. With standard regret loss, CLUTR shows a lower regret value, but shows similar pattern. The CLUTR agent achieves better returns with flexible loss throughout the training.

Figure 13 compares the mean regret and agent training returns with PAIRED. CLUTR with standard loss shows much lower regret than PAIRED (Figure 13a). Figure 13b shows that the CLUTR agents compete closely, while PAIRED antagonist achieves much higher returns than the PAIRED agent which leads to higher regret returns for the teacher agent but results in a weak student agent. To test the Zero-shot generalization, we evaluate CLUTR with the standard loss on the full F1 benchmark. Figure 14 shows CLUTR with standard regret loss outperforms PAIRED in all the 20 test tracks. This implies that CLUTR outperforms PAIRED irrespective of the choice of the loss function (standard/flexible). Figure 15 compares the sample efficiency of CLUTR with the standard regret loss with PAIRED by evaluating the agents on four selected tracks (Vanilla, Singapore, Germany, Italy) during training. It can be seen that CLUTR, even without the regret loss, outperforms PAIRED significantly. We note that these test environments were not used in any way, neither during training CLUTR (and PAIRED) nor while designing it.

As mentioned in Jiang et al. (2021a) PAIRED overexploits the relative strengths of the antagonist over the protagonist and generates a curriculum that gradually reduces the task complexity. However, CLUTR overcomes this and generates a curriculum where the agent and the antagonist closely compete (Figure 13b) and shows a robust generalization on the unseen F1 benchmark.

## D.3 EXTENDED ANALYSIS ON IMPACT OF SORTING TRAINING DATA FOR VAE TRAINING

The non-sorted dataset was generated by shuffling each track of the original VAE training dataset 10 different times, resulting in a 10X bigger dataset (10M tracks). It was trained for 5X longer for 5M training steps. We planned on training for 10M gradient steps (10X than the original VAE) but stopped at 5M as it converged much sooner. We ran both CLUTR and CLUTR-shuffled, i.e.,

| Track | DR | PLR | Robust PLR | PAIRED | REPAIRED | CLUTR (2M) | Attention Agent |
|---|---|---|---|---|---|---|---|
| Australia | 484 ± 29 | 545 ± 23 | **692 ± 15** | 100 ± 22 | 414 ± 27 | **683 ± 20** | 826 |
| Austria | 409 ± 21 | 442 ± 18 | **615 ± 13** | 92 ± 24 | 345 ± 19 | 507 ± 19 | *511* |
| Bahrain | 298 ± 27 | 411 ± 22 | **590 ± 15** | -35 ± 19 | 295 ± 23 | 414 ± 20 | *372* |
| Belgium | 328 ± 16 | 327 ± 15 | **474 ± 12** | 72 ± 20 | 293 ± 19 | 429 ± 15 | 668 |
| Brazil | 309 ± 23 | 387 ± 17 | **455 ± 13** | 76 ± 18 | 256 ± 19 | 363 ± 18 | *145* |
| China | 115 ± 24 | 84 ± 20 | **228 ± 24** | -101 ± 9 | 7 ± 18 | **254 ± 28** | 344 |
| France | 279 ± 32 | 290 ± 35 | **478 ± 22** | -81 ± 13 | 240 ± 29 | **498 ± 31** | *153* |
| Germany | 274 ± 23 | 388 ± 20 | **499 ± 18** | -33 ± 16 | 272 ± 22 | 404 ± 20 | *214* |
| Hungary | 465 ± 32 | 533 ± 26 | **708 ± 17** | 98 ± 29 | 414 ± 29 | 630 ± 24 | 769 |
| Italy | 461 ± 27 | 588 ± 20 | **625 ± 12** | 132 ± 24 | 371 ± 25 | **639 ± 16** | 798 |
| Malaysia | 236 ± 25 | 283 ± 20 | **400 ± 18** | -26 ± 17 | 200 ± 17 | **426 ± 22** | *300* |
| Mexico | 458 ± 33 | 561 ± 21 | **712 ± 12** | 67 ± 31 | 415 ± 30 | 627 ± 19 | *580* |
| Monaco | 268 ± 28 | 360 ± 32 | **486 ± 19** | -28 ± 18 | 256 ± 26 | **460 ± 29** | 835 |
| Netherlands | 328 ± 26 | 418 ± 21 | 419 ± 25 | 70 ± 20 | 307 ± 21 | **488 ± 21** | *131* |
| Portugal | 324 ± 27 | 407 ± 15 | **483 ± 13** | -49 ± 13 | 265 ± 21 | **462 ± 20** | 606 |
| Russia | 382 ± 30 | 479 ± 24 | **649 ± 14** | 51 ± 21 | 419 ± 25 | 497 ± 23 | 732 |
| Singapore | 336 ± 29 | 386 ± 22 | **566 ± 15** | -35 ± 14 | 274 ± 21 | 382 ± 19 | *276* |
| Spain | 433 ± 24 | 482 ± 17 | **622 ± 14** | 134 ± 24 | 358 ± 24 | 496 ± 15 | 759 |
| UK | 393 ± 28 | 456 ± 16 | **538 ± 17** | 138 ± 25 | 380 ± 22 | 471 ± 19 | 729 |
| USA | 263 ± 31 | 243 ± 28 | **381 ± 33** | -119 ± 11 | 120 ± 25 | 238 ± 31 | *-192* |
| Mean | 342 ± 27 | 404 ± 22 | **531 ± 17** | 26 ± 19 | 295 ± 23 | 468 ± 21 | *478* |

Table 4: Comparison between CLUTR and other UED algorithms. Boldface denotes SOTA among UED algorithms, while italic in the Attention Agent colum means, CLUTR is comparable/outperforms the attention agent on that track. CLUTR outperforms PAIRED by a big margin with 18x bigger mean return on the entire F1 Dataset. CLUTR also outperforms Domain Randomization, PLR, and REPAIRED and only falls short to Robust PLR. Nonetheless, CLUTR shows competitive results compared to Robust PLR, showing comparable results in seven out of the 20 test tracks and outperforming in the Netherlands track. CLUTR also outperforms the non-UED SOTA on the full F1 dataset. CLUTR outperforms the Attention Agent on 9 out of the 20 tracks and shows comparableperformance in one. It must be noted, all the UED methods except CLUTR was trained for 5M timesteps where CLUTR was run for 2M timesteps.

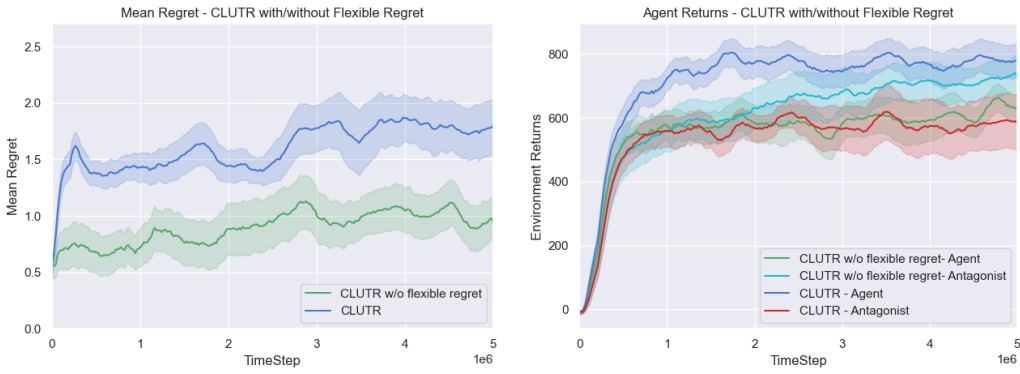

(a) Mean Regret - Car Racing - with vs without flexible regret loss

(b) Returns on UED generated Car Racing tracks - with vs without flexible regret loss

Figure 12: Mean Regret and agent returns during training CLUTR (with flexible regret) vs CLUTR with standard PAIRED regret approximation.

CLUTR with a VAE trained on non-sorted data up to 5M timesteps. CLUTR-shuffled shows inferior performance and also signs of unlearning compared to CLUTR. Figure 16 shows detailed experiment results.

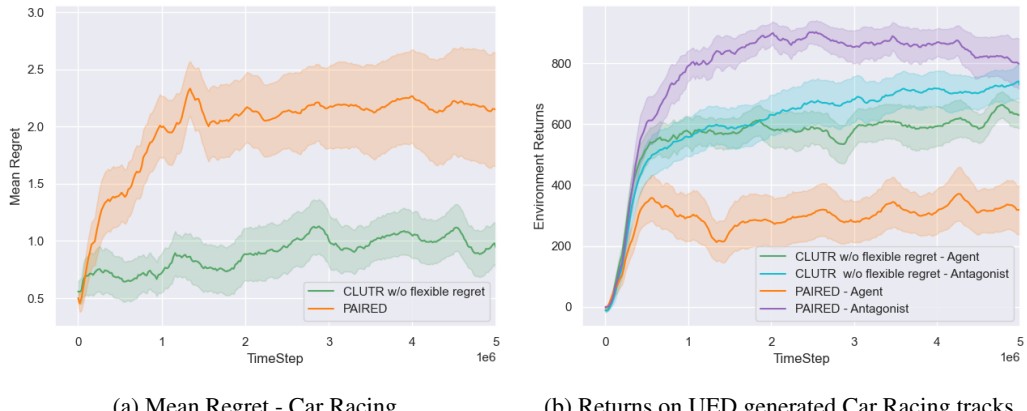

(a) Mean Regret - Car Racing  (b) Returns on UED generated Car Racing tracks

Figure 13: Mean Regret and agent returns during training CLUTR with standard PAIRED regret loss (i.e., without the flexible regret). CLUTR shows a smaller regret value(i.e., closely competing agent and antagonist), indicating a better UED curriculum.

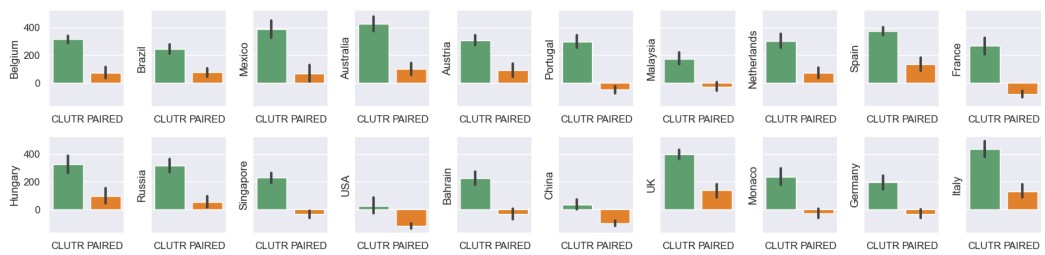

Figure 14: Zero-shot generalization of both PAIRED and CLUTR (with the standard regret loss) agents after 5M timesteps on the full F1 benchmark. CLUTR with the standard regret loss outperforms PAIRED on every track. For each track, we test the agents on 10 different episodes and the error bar denotes the standard error.

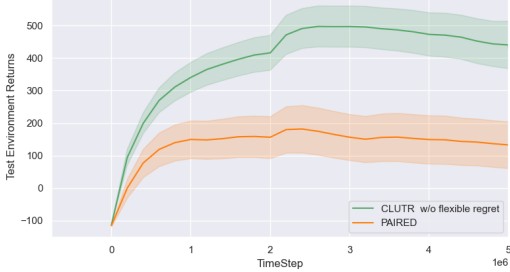

Figure 15: Test Returns on Selected Tracks (Vanilla, Singapore, Germany, and Italy) of CLUTR with standard PAIRED regret loss alongside PAIRED performance.

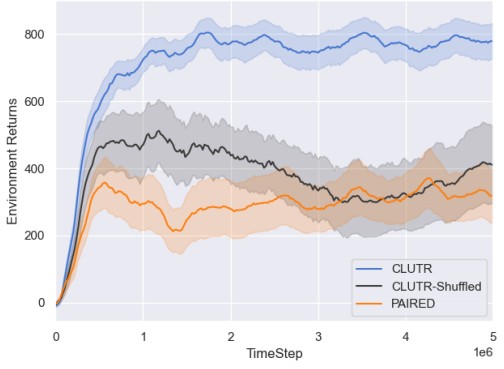 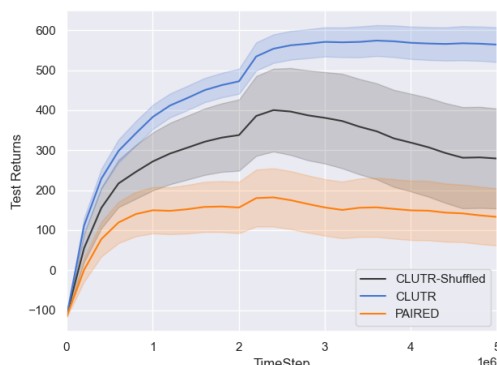

(a) During training CLUTR agent achieves higher returns while, CLUTR-shuffled agent shows lower returns. CLUTR-Shuffled agent's return is also less stable showing a decrease and increase.

(b) CLUTR achieves higher and more stable mean returns on the selected tracks. CLUTR-Shuffle shows signs of unlearning.

Figure 16: Analysis of sorting training data for VAE. Trained on shuffled data, CLUTR-Shuffled performs inferior compared to CLUTR and shows signs of unlearning.

### D.4 Impact of Task Representation Learning

In this section, we discuss the impact of the learned task representation on performance. In Section 5.4, we showed that if we finetune the VAE decoder during curriculum learning, the overall performance drops significantly (Figure 6). To get a better understanding, in Figure 17, we plot how much the performance deviates as the VAE decoder changes during the training process. The curve in red shows the deviation of the decoder from its pretrained weights as it is fine-tuned during the training. We estimate the deviation as the L2 distance between the finetuned and the pretrained decoder weights. The green curve shows the performance drop from CLUTR (with standard loss). To estimate the performance drop, we periodically evaluate both CLUTR and CLUTR with Finetuned VAE, on the selected test tracks during training. From the figure, we observe that,

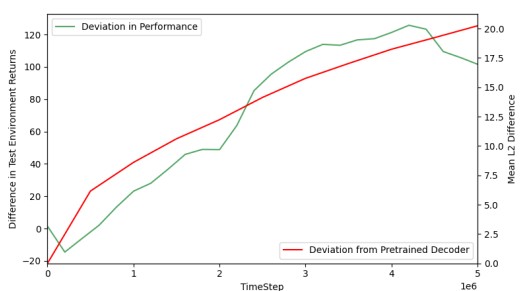

Figure 17: Impact of pretrained decoder weights on performance. The red curve plots the deviation of the decoder from its pretrained weights as it is finetuned. The green curve shows the performance drop from CLUTR with the standard loss. These curves suggest that pretrained weights are crucial for performance.

as the decoder weights are finetuned, they become increasingly different from the initial pretrained weights. At the same time, the overall performance gap from CLUTR also increases. This suggests that the pretrained VAE weights are crucial for better performance.

Furthermore, the quality of the learned representation depends on the quality of the data they are trained on. In section 5.5, we showed that a VAE trained on a non-sorted dataset significantly deteriorates the performance (Figure 6). This further suggests that the learned representation has a significant impact on performance. We also want to note that both of these variations (CLUTR with Finetuned VAE and the CLUTR with Shuffled VAE) perform much better than PAIRED, which suggests that, though CLUTR's performance depends on the representation, with a reasonable representation, it can still perform better than PAIRED.

# E    DETAILED RESULTS ON MINIGRID

## E.1    CURRICULUM ANALYSIS

Figure 19 shows 3D Histograms showing the frequency of the generated grids against the total number of obstacles they contain. PAIRED starts with a high number of obstacles and then degenerates quickly into grids with very few numbers of obstacles and stays similar for a significant number of steps. Eventually, the number of obstacles increases sharply, converging into a band of around 20 to 40 obstacles on average. On the other hand, in CLUTR, the number of obstacles starts flat, centers around a peak around the middle but still with a wide interval for some number of steps, and the peak drops slightly while the interval stays almost the same. After the 'convergence', PAIRED rarely generates grids with fewer or more obstacles than the band it converges to. On the contrary, CLUTR still generates grids with few or many blocks, which might help to address unlearning or improve the agents on grids with more obstacles, respectively. The above observations illustrate that we can achieve a more efficient curriculum learning without making the problem too easy early or without focusing on a narrow interval with a flat distribution later. Instead, we can start with a wide interval and gradually focus on a peak around the middle without making the interval very narrow.

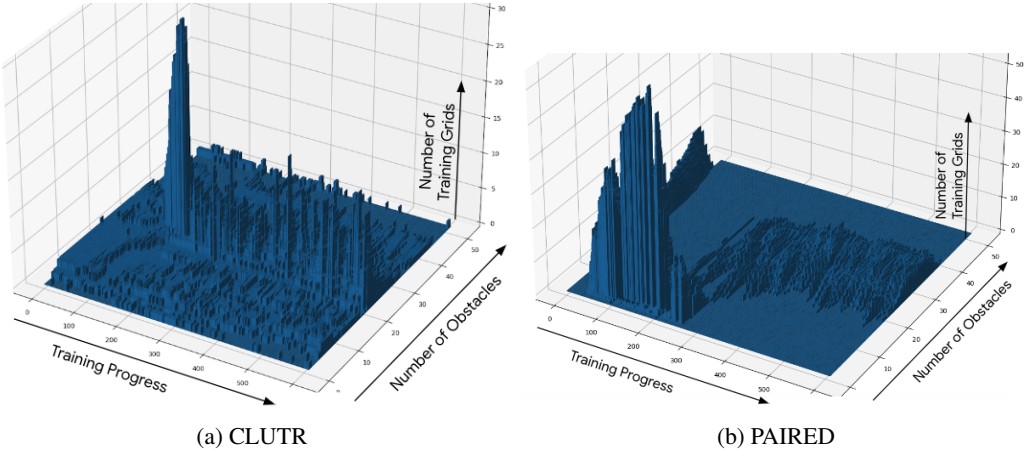

|          (a) CLUTR          |          (b) PAIRED          |

Figure 18: 3D Histograms showing the frequency of the generated grids against the total number of blocks they contain. Both PAIRED and CLUTR converge to a similar band of grids. However, CLUTR converges much faster.

Figure 19a shows the average episode lengths of both CLUTR and PAIRED. The curves show both methods start with long episodes—indicating at the beginning, the agents do not solve the training grids consistently, and many of the episodes end due to timeout. As the agents learn, the episodes become shorter for both methods until they converge to a small value. However, CLUTR converges sooner than PAIRED.

We also compare the average solution length of the solved training grids. Both PAIRED and CLUTR show a similar pattern. However, PAIRED converges to a larger value than CLUTR. This might indicate that CLUTR is solving the environments more efficiently. This might also mean that CLUTR is solving some easier tasks (e.g., fewer obstacles, as we noticed from Figure 19) even after convergence lowering its average solved path length slightly.

## E.2    CLUTR CURRICULUM VS. DOMAIN RANDOMIZED CURRICULUM ON THE LATENT SPACE: DOES CLUTR TEACHER DEGENERATES INTO A RANDOMIZED POLICY?

To answer whether CLUTR teacher actually learns something or degenerates into a randomized policy, we compare the curriculum generated by CLUTR with a random uniform (i.e., Domain Randomization) curriculum. We generate the DR curriculum by repeatedly sampling the trained VAE (the same VAE used by CLUTR) with a uniform random distribution. Figure 20 shows the comparison characterizing the grids by the number of obstacles they contain similarly as the previous section. As expected, we can see that the DR curriculum generates grids with obstacles ranging from

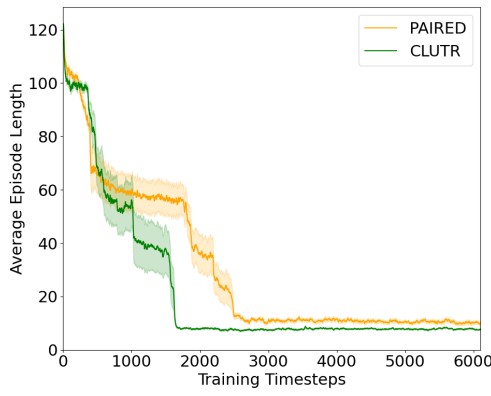 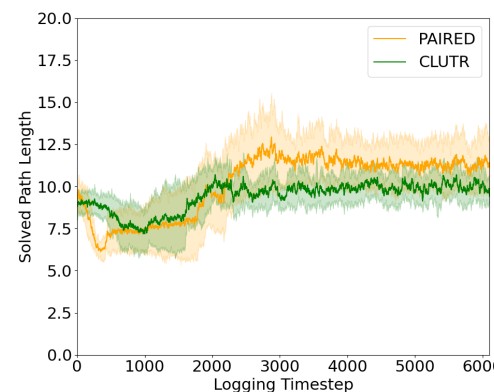

(a) Average length of the training episodes. CLUTR converges sooner than PAIRED to a shorter episode length.

(b) Average solution length of the solved training tasks.

Figure 19: Comparison of CLUTR and PAIRED curriculum based on properties of the generated grids.

0 to 50. The histograms clearly visualize the significant differences in the curricula, implying the CLUTR teacher indeed learns a useful curriculum as suggested by the empirical result.

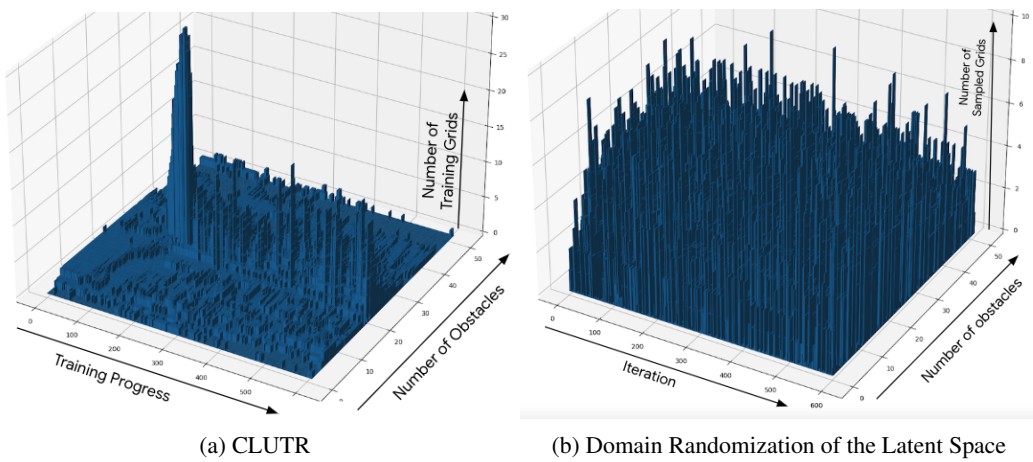

(a) CLUTR

(b) Domain Randomization of the Latent Space

Figure 20: 3D Histograms showing the frequency of the CLUTR generated grids against the total number of blocks they contain vs. randomly generated grids.

### E.3 ANALYSIS OF THE LATENT TASK MANIFOLD

To grow a sense of the latent task manifold, we linearly interpolate in the latent space between an empty grid and a 15x15 version of the FourRoom grid (shown in Figure 21). Figure 22 visualizes the interpolation results. We first get the latent vectors of the empty grid and the target FourRoom task using the VAE encoder. We then linearly interpolate 23 equidistant points between them. At last, we reconstruct the grids from these vectors using our decoder. From Figure 22 we see that, as we interpolate in the latent space, the reconstructed grid incrementally adds more obstacles and the grids start to look more like the FourRoom target grid. We note that the reconstruction is not perfect. We also note that the increase in the number of obstacles is not uniform, e.g., the first 5 reconstructed grids are all empty grids, and more obstacles are added near the target

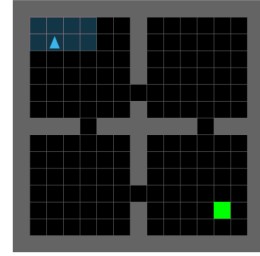

Figure 21: 15X15 Four-Rooms

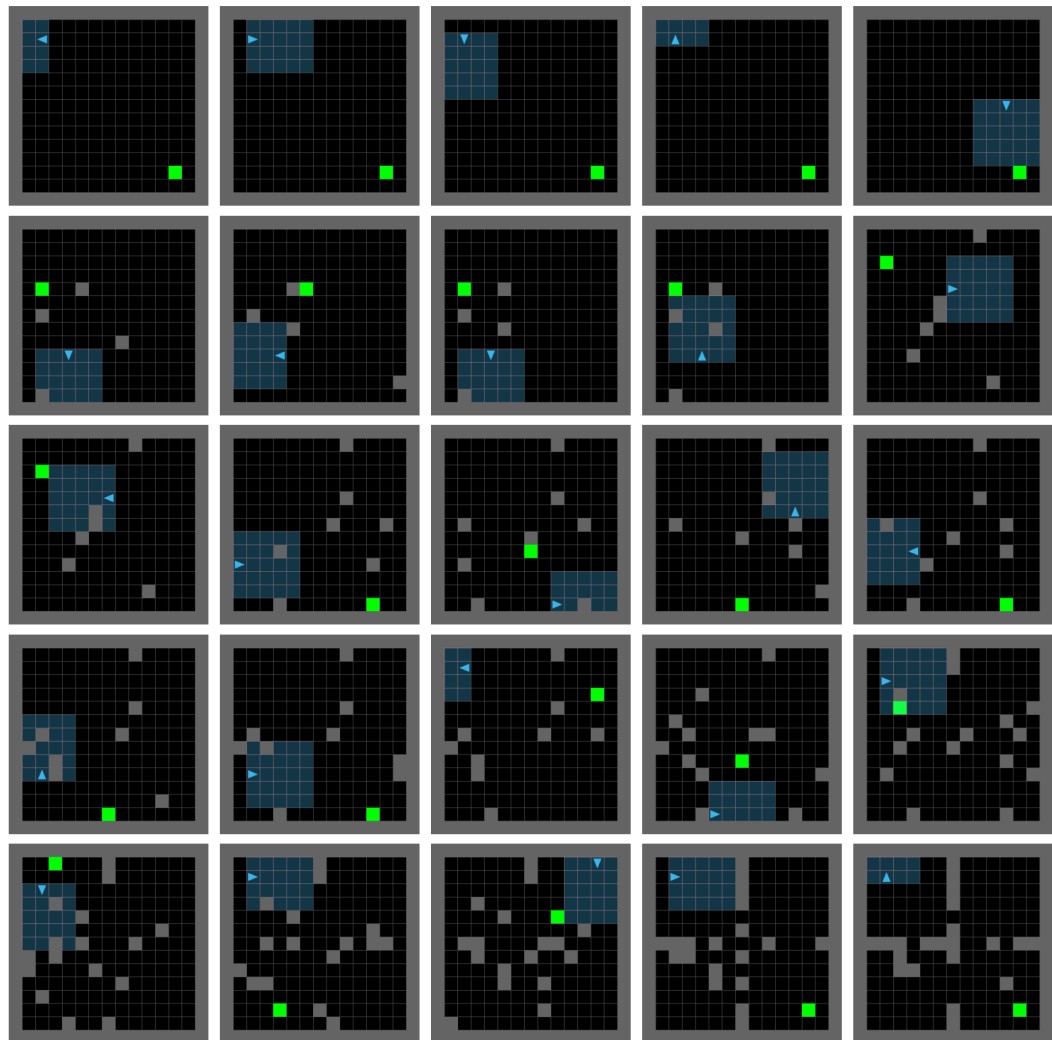

Figure 22: A linear interpolation between an empty grid and 15x15 version of the Four-Room grid (Figure 21) in the latent space. The grids are organized from top-left to bottom-right in row-major order.

point. Overall, this experiment provides an insight that the latent space holds a useful structure, which CLUTR teacher utilizes to generate the curriculum.

