# OpenReview forum: "CLUTR: Curriculum Learning via Unsupervised Task Representation Learning"
_ICLR.cc/2023/Conference — Submitted to ICLR 2023_

### Official Review · Reviewer_eq4e · 2022-10-19

**Confidence:** 5
**Correctness:** 3
**Technical Novelty And Significance:** 2
**Empirical Novelty And Significance:** 2
**Recommendation:** 5

**Clarity, Quality, Novelty And Reproducibility:**

### **Clarity**
There are some clarification problems (as mentioned in the weakness part) that need to be solved. Although the general idea of the proposed method is easy to understand, a lot of details are missing.

### **Quality**
The quality of this paper is limited by the presentation. A clearer organization would improve the quality.

### **Novelty**
The novelty of this paper is ok but maybe around the average level. If I understand correctly, the only modification of this paper is replacing the task generator with a pre-trained VAE.

### **Reproducibility**
Not sure. Code is not provided.




**Strength And Weaknesses:**

### **Strength**

* **Interesting topic.** Unsupervised Environment Design is an important direction in curriculum reinforcement learning. Previous works usually suffer from unstable training problems. This paper targets at solving this problem by using the decoder of VAE to generate tasks.

* **Comprehensive experiment.** The experiment part looks good to me. Some important factors are explored respectively.

### **Weaknesses**

* **Things to be clarified.**
    * The proposed method heavily depends on PAIRED but details of PAIRED are missing in Section 4.3.
    * In the 4th line of Algorithm 1, what does it mean by “Use adversary to sample latent task vector”? If the adversary is a parametrized model, what distribution is used for sampling z?
    * What is single-step RL? It seems not a widely used term but the definition is missing. I guess the authors mean that their method does not do sequential generation with the interaction with the environment.
    * The second difference between CLUTR and PAIRED is hard to follow. More explanation is needed. Why does the state space of PAIRED rely on the underlying POMDP?
    * No example of permutation parameters is given in the experiment environment. The authors say that “for a navigation task, a set of obstacles corresponds to factorially different permutations of the parameters.” What is the obstacle in the Car racing experiment?
    * I don’t think VAE(z, E) in equation (1) is a standard notation. It should be clarified.


* **Formulation of CLUTR.** The derivation in Appendix B seems redundant to me. According to Figure 1, variable R actually does not involve the VAE inference since it only depends on another observable variable E. Therefore, Equation (1) is just the ELBO of VAE plus an arbitrary regularization term that relates R and E. I think the authors can directly say that rather than taking a detour.

* **Motivation.** If I understand correctly, this paper proposes to use VAE to generate tasks because of two advantages, i.e., long-horizon credit assignment and permutation invariant. However, searching in the latent space itself is not an easy task since the smoothness of latent space heavily depends on the training and quality of the dataset. So, does this method always outperform PAIRED in general? For example, when most of the uniformly sampled environment parameters are invalid.


**Summary Of The Paper:**

This paper targets the topic of unsupervised environment design (UED) in curriculum reinforcement learning. In particular, it modifies PAIRED by using the decoder of a VAE as the task generator. With this improvement, the model decouples task representation and curriculum learning into a two-stage optimization. Thus, the authors observe that the training is more stable and the final reward is higher than existing UED methods.

**Summary Of The Review:**

Due to my concerns about the clarification, I suggest rejecting this paper. The authors are welcome to address my questions.

---

> ### Author Response · Authors · 2022-11-18
> **Response to Reviewer eq4e (2)**
>
> **Weakness - Motivation:**
>
> **Difficulty of Learning to Search in a Latent Space:**
>
> Thank you for raising this important point. While (1) the introduction of a latent space provides CLUTR the advantage of solving long-term credit assignment and permutation invariance faced by the other parameter-space adaptive teachers (i.e., PAIRED & REPAIRED), (2) searching in the latent space is not easy, and the quality of the latent space depends on the training and quality of the dataset. Hence, the success of CLUTR is determined by if the benefits of (1) outweigh the challenges of (2).
>
> Now, our empirical results strongly suggest that CLUTR outperforms PAIRED and REPAIRED. This indicates that the introduction of latent space indeed can improve adaptive-teacher UED algorithms and the benefits of introducing the latent space outweigh the challenges it introduces. Furthermore, CLUTR uses randomly generated datasets to train the VAE and can still generate effective curriculums, which shows the strength of the proposed approach. We believe our work will inspire further research on unsupervised latent space curriculum design.
>
>
> **Does CLUTR always outperform PAIRED?:**
>
> Theoretically, CLUTR holds the same robustness guarantees as PAIRED. However, both PAIRED and CLUTR make a few approximations and practical choices (e.g., regret approximation as measuring the true regret is not feasible). Additionally, the robustness guarantee (of both PAIRED and CLUTR) depends on reaching the Nash equilibrium of the multiagent adversarial game. However, gradient-based multi-agent RL has no convergence guarantees and often fails to converge in practice [1]. Hence in practice, the algorithms deviate from the theoretical guarantees.
>
>
> In practice, we evaluated CLUTR in the environments which are typically used in this domain and showed it performs better both in terms of sample efficiency and zero-shot generalization. The reviewer raised an interesting question about whether CLUTR would always outperform PAIRED. We make no such claims. It would be an interesting future direction to investigate under which conditions a latent-space teacher outperforms a parameter-space teacher and vice-versa. We amended the paper to better reflect this point (i.e., Conclusion and Appendix B.2).
>
>
>
> [1] Eric Mazumdar, Lillian J. Ratliff, Michael I. Jordan, and S. Shankar Sastry. Policy-gradient algorithms have no guarantees of convergence in linear quadratic games, 2019. URL https://arxiv.org/abs/1907.03712.
>
>
>
> **Clarity & Quality of Presentation:** We tried to address the concerns raised by the reviewer to our best.
>
>
>
> **Novelty:**
>
> Training a teacher with RL is one of the most critical challenges of parameter-space adaptive-teacher UED algorithms, i.e., PAIRED, REPAIRED [1], and often regarded as the Achilles’ heel of adaptive-teacher UEDs [2]. CLUTR makes an important step towards solving this problem by introducing latent space teachers—a novel contribution in this domain. We conducted extensive experiments to evaluate the strengths and drawbacks of our approach. Our experimental results and analysis strongly support the advantage of our approach.
>
> Furthermore, our approach also draws motivation from the recent successful trend of using pretrained networks in domains such as, NLP, Vision, and RL. Our use of a pretrained network and VAE is also novel in its context.
>
> Additionally, the ability to design a useful latent space in an unsupervised manner from a randomly generated dataset without any costly interaction with the simulator is very useful.
>
>
>   [1] Jiang, Minqi, et al. "Replay-guided adversarial environment design." Advances in Neural Information Processing Systems 34 (2021): 1884-1897.
>
>
> [2] Parker-Holder, Jack, et al. "Evolving Curricula with Regret-Based Environment Design." arXiv preprint arXiv:2203.01302 (2022).
>
> **Reproducibility:** We have shared our code, training data, and models here: [https://github.com/clutr/clutr](https://github.com/clutr/clutr)

---

> ### Author Response · Authors · 2022-11-18
> **Response to Reviewer eq4e**
>
> We thank the reviewer for their valuable feedback. We appreciate their support of the importance of our work and comprehensive experiments. We hope our clarifications and the revised manuscript would address the raised concerns about clarity and motivation.
>
> **Weaknesses - Clarification:**
>
>
>
> **Details of PAIRED in Section 4.3:** Thank you for the feedback. CLUTR outline is very similar to PAIRED, differing only in the first two lines of the main loop in Algorithm 1 to incorporate the latent space. We have revised Section 4.3 to incorporate the feedback.
>
> **Confusion about sampling latent vectors in Algorithm 1:** Thank you for pointing this out. The latent vectors are the actions generated by the teacher agent. In our implementation, we used Beta distribution to sample actions for CarRacing and Categorical for Minigrid teachers. These details are discussed in Section C.2 of the Appendix (Teacher Architecture). For better readability and to avoid confusion, we have revised the corresponding line of the algorithm.
>
>
> **Single-step RL:** Thanks for the feedback. We replaced this with “contextual bandit” in the paper.
>
> **Difficulty understanding the Second Difference between PAIRED and CLUTR:** PAIRED-variants typically observe the state of the partially generated task to generate the next parameter. Hence depending on the state space, they require designing different teacher architectures for different environments, e.g., for Minigrid the PAIRED teacher receives the partially generated grids as observation and uses a Convolutional Layer to process it. CLUTR teacher architecture, however, is agnostic of the problem domain, i.e, it receives a flat latent vector. Hence the same architecture can be used across different problems. We have revised this point in our revised paper for better readability.
>
> **Permutation Example:** We have added an example in the main text for better understanding: “Consider a 13x13 grid for a navigation task, where the locations are numbered from 1 to 169. Also consider a wall made of four obstacles spanning the locations: {21, 22, 23, 24}. This wall can be represented using any permutation of this set, e.g., {22, 24, 23, 21}, {23, 21, 24, 22}, resulting in a combinatorial explosion.”
>
> For CarRacing, the parameters do not denote obstacles. The CarRacing tasks are racing tracks defined using bezier curves. The parameters define the control points for the bezier curve. The parameterization is discussed in Section 5.1, and Section C.1 of the appendix contains further details.
>
>
> **VAE(z, E) Notation:** Thank you. We have added a description of the term in the paper for clarification.
>
>
>
> **Weaknesses - CLUTR Formulation:**
>
> Thanks for the feedback. We revised Section 4.1 to incorporate this feedback. We clarified in the main text that our objective is VAE + some regularization and left the appendix for details on how such an objective can be derived using a principled variational approach to avoid redundancy.

---

### Official Review · Reviewer_Ui8Z · 2022-10-23

**Confidence:** 2
**Correctness:** 4
**Technical Novelty And Significance:** 2
**Empirical Novelty And Significance:** 3
**Recommendation:** 5

**Clarity, Quality, Novelty And Reproducibility:**

In general, the paper is well presented and well written. The authors claim they will release the code upon acceptance.


**Strength And Weaknesses:**


strength: The paper is well written.
The experiments showed promising results and covered a range of hypothesis claimed in the paper.


weakness: the novalty of the paper is somewhat limited. The main techniques used in CLUTR are based on VAE and PAIRED, which are existing methods.

I am curious to know the effect of learning the representation of the tasks.
For example, how sensitive does the performance depend on the fitting of the VAE representation? Is the VAE represntation is mis-specified, how much does it worsen the performance?


Minor:
* E is not defined in 4.1
* equations in 4.1 are better written with $\log q(z)$
* missing space in section 4: Section4.4 -> Section 4.4


**Summary Of The Paper:**

This paper proposes CLUTR, a curriculum learning algorithm based on PAIRED, which decouples
task representation and curriculum learning. The task representation is learned
using a LSTM-based recurrent VAE, after which the teacher updates the curriculum and
the protagonist and antagonist policies are updated according to the regret
of the generated tasks. The experiments showed empirically that CLUTR outperforms
state-of-the-art UED methods in terms of sample efficientyc and generalization
 on a range of tasks, and performs compratably to the non-UED state-of-the-art on the CarRacing task.


**Summary Of The Review:**

The paper tackles the problem of sample inefficiency in RL problems by pre-training a VAE to embed the tasks in to a manifold and update policies based on sampled trajectories from the tasks, however, the novelty of the paper is a bit limited. Having said that. given the empirical performance on the extensive sets of experiments, the proposed method may be helpful for the community in practice.

---

> ### Author Response · Authors · 2022-11-18
> **Response to Reviewer Ui8Z**
>
>
> We thank the reviewer for their valuable feedback. We appreciate their support of our writing, empirical results, and analysis. We hope our clarifications and the revised manuscript would address the raised concerns about the novelty of the method.
>
>
> **Concern on Novelty:**
>
> We are using existing techniques, VAE and PAIRED, *in a novel way to* solve an important problem in this domain.
>
> 1.  VAEs have been widely used in the literature. However, the explicit use of latent task manifold in UEDs to generate a curriculum is novel, and so is the use of VAE. Furthermore, the ability to design a useful latent space in an unsupervised manner from a randomly generated dataset without any costly interaction with the simulator is very useful and novel.
>
>
> 2.  Training a teacher with RL is one of the most critical challenges of parameter-space adaptive-teacher UED algorithms, i.e., PAIRED, REPAIRED [1], and often regarded as the Achilles’ heel of adaptive-teacher UEDs [2]. CLUTR makes an important step towards solving this problem by introducing latent space teachers—a novel contribution in this domain. We conducted extensive experiments to evaluate the strengths and drawbacks of our approach., Our experimental results and analysis strongly support the advantage of our approach.
>
>
> 3.  Our approach also draws motivation from the recent successful trend of using pretrained networks in domains such as NLP, Vision, and RL. Our use of a pretrained network is also novel in its context.
>
>
> We hope the above points address the concern of the reviewer regarding the novelty of our method.
>
>
>
>
> [1] Jiang, Minqi, et al. "Replay-guided adversarial environment design." Advances in Neural Information Processing Systems 34 (2021): 1884-1897.
>
> [2] Parker-Holder, Jack, et al. "Evolving Curricula with Regret-Based Environment Design." arXiv preprint arXiv:2203.01302 (2022).
>
>
>
>
> **Effect of learning the task representation:**
>
> The task representations are crucial to our performance. We added a new analysis (Appendix D.4) to show that, when we allow for finetuning the VAE during curriculum learning (CLUTR - Finetuned VAE, also discussed in Section 5.4), as the training progresses, the VAE becomes increasingly different from the initial pretrained weights. At the same time, the drop in performance (from CLUTR) increases too. This suggests that pretrained VAE weights are crucial for better performance. Furthermore, in Section 5.5, we show that pretraining the VAE using a non-sorted dataset (CLUTR - Shuffled VAE) also deteriorates the performance, indicating the impact of learned representation on performance. We also note that CLUTR - Finetuned VAE and CLUTR - Shuffled VAE perform significantly better than PAIRED. This suggests that even though CLUTR performance is sensitive to the quality of task representation, it can still outperform PAIRED given a reasonable representation. We have added a new Section D.4, discussing the effect of learning the task representation in detail.
>
>
>
>
>
> **Other Minor Concerns:**
>
>
> -   Definition of E: E is a random variable denoting the environment. We updated the manuscript.
>
> -   Missing Space: Thank you for pointing that out. We corrected it in the updated manuscript.
>
>
>
>
>
> **Reproducibility:**
>
>
> We have shared our code, training data, and models here: [https://github.com/clutr/clutr](https://github.com/clutr/clutr)

---

### Official Review · Reviewer_NPYm · 2022-10-23

**Confidence:** 4
**Correctness:** 1
**Technical Novelty And Significance:** 3
**Empirical Novelty And Significance:** Not applicable
**Recommendation:** 3

**Clarity, Quality, Novelty And Reproducibility:**

### Clarity
The paper does a good job of explaining the high-level details of the method, as well as providing background on related topics and the experimental setup. However, there are several important details that are missing:
- As previously mentioned, it is not clear whether the main results are based on running CLUTR with the flexible regret objective or the standard PAIRED regret objective.
- The maze navigation results should report the number of training seeds used.
- It is not clear why the CarRacing experiments compare to the additional Robust PLR and ACCEL baselines, but the navigation experiments do not.
- In Section 5.4, the method of fine-tuning the decoder throughout training on the agent's regret is not provided anywhere in the paper. Could the authors provide details on how this fine-tuning is performed? It is unclear, as it seems the VAE was not trained to predict the regret values—which are not available during pre-training.

### Quality
- Overall, the paper has several rough edges and could benefit from copy editing.
- There are several claims that are incorrect or unjustified by experimental evidence, which are detailed in the Weaknesses section of this review.

### Novelty
- The idea of latent-space task design has been explored in other works such as Florensa et al, 2017, which the authors should also cite.
- The idea of learning a task representation for UED has not been previously explored in detail.

### Reproducibility
- Due to the lack of clarity around key details described in this review, and the lack of experimental code from the authors, the results of this study are not reproducible based on the information available.

**References**
Florensa, Carlos, et al. "Automatic goal generation for reinforcement learning agents." International conference on machine learning. PMLR, 2018.

**Strength And Weaknesses:**

### Strengths
- The benefits of performing UED on a learned latent manifold of the task space is well-motivated.
- The connection between CLUTR and prior works is clearly discussed.
- The CarRacing experiments compare against a comprehensive set of baselines.

### Weaknesses
Despite what seem like promising results showing CLUTR outperforms PAIRED, there are several important points that are left unclear and claims that seem to be incorrect:

**The regret objective used by CLUTR is unclear**

- It is not clear whether CLUTR uses the flexible PAIRED objective or the standard PAIRED objective. Appendix D.2 seeks to compare CLUTR with the _standard PAIRED regret objective_, implying the main results use the flexible PAIRED objective.
- If CLUTR uses the flexible regret objective, then the main experiment results should compare to PAIRED with the flexible regret objective, also known as "flexible PAIRED" in Dennis et al, 2020. It must be shown that CLUTR's gains over PAIRED with the standard regret objective are not in fact due to the choice of regret objective, rather than the task representation, as claimed.
- If CLUTR does in fact use the flexible regret objective, then Theorem 1 from Dennis et al, 2020, which guarantees a minimax-regret policy for the protagonist at Nash equilibrium of the PAIRED game, is no longer directly applicable—since the flexible regret objective changes the nature of the underlying game between the three agents.
- Moreover, the flexible regret objective was not introduced in Gur et al, 2021 as stated in the first paragraph of Section 3.2, but clearly detailed and evaluated in the appendix of Dennis et al, 2020.

**CLUTR does not have access to the full task space**

While PAIRED, in principle, has the ability to generate any 50-block maze, this is not necessarily true for CLUTR. This is because CLUTR is trained on a limited subset of all possible tasks, and the learned latent space is not guaranteed to contain all possible mazes. Therefore, Property 3 of CLUTR, stated on Page 5, is incorrect.

**Interpretation of Figure 4 and 5 is flawed**

- In Figure 5, CLUTR's adversary achieves lower regret than that of PAIRED. This result implies that the latent task representation space makes the optimization of the regret objective a harder problem for the teacher, rather than easier, as claimed in the paper.
- Figure 4 shows the solved rate of the agent on the _training_ levels proposed by the CLUTR and PAIRED adversaries. Since the goal of UED is to produce an adversarial curriculum, it seems that the higher solved rates achieved by the agent under the CLUTR adversary implies the CLUTR adversary is less effective and does a worse job at optimizing for agent's regret.

**The results do not provide evidence for whether the VAE and CLUTR adversary learn anything**

- While the paper argues that CLUTR has the advantage of avoiding sequential credit-assignment, CLUTR must pay the additional cost of a much larger action space—based on the latent-space dimensionality. Thus, CLUTR may in fact face a much more difficult learning problem than PAIRED. Further, the VAE must also learn a useful representation of the task space, the success of which is not shown in the paper.
- The paper should provide visualizations of how tasks are distributed within the VAE's latent space to provide evidence that meaningful latent structure is being learned via their method. The training curves for the VAE would also be useful to see.
- The paper should provide evidence that CLUTR's metrics in Figures 3, 4, and 5 are different from that of domain randomization—which would show that CLUTR's adversary is learning to exploit the structure in the latent space, rather than resorting to a randomized policy.

These figures should compare to domain randomization (DR), and importantly, show that CLUTR is doing something meaningfully different from randomizing over the latent space (which would be case if the adversary policy has difficulty learning to design in this space).

**Curriculum analysis is lacking**

- The main emergent complexity and curriculum results can benefit from thorough analysis. In particular, the authors should plot the solution path length and number of blocks in training maze tasks proposed by the CLUTR adversary, in comparison to other methods.
- Since the CLUTR tasks are generated by decoding latent representations using the pre-trained VAE, it is possible that CLUTR produces mazes with more obstacle blocks than PAIRED, which is strictly limited to at most 50 blocks. Prior works (Jiang et al, 2021) show that going from 25 to 50 blocks improves the OOD transfer performance of all methods compared, so it seems that this factor is a confounder.
- Importantly, this curriculum analysis should compare to DR to show that CLUTR learns a meaningful curriculum.

**Missing several key details**

In addition to the lack of clarity around CLUTR's regret objective, a few other important details seem to be missing. (See the Clarity section for details.)


**Summary Of The Paper:**

This paper investigates learning a task representation space for use in unsupervised environment design (UED), with an algorithm called PAIRED. PAIRED trains an RL agent, the adversary, to configure the environment parameters throughout training, in tandem with two RL agents that learn to solve these configurations (each configuration is termed a "task"). The adversary is allied with one of the agents, called the antagonist, and seeks to maximize the margin of return by which the antagonist outperforms the other agent, called the protagonist. Thus, PAIRED produces a curriculum that maximizes a lower bound on the regret experienced by the protagonist, leading to a robust minimax-regret policy at equilibrium. This paper then proposes CLUTR, a method that first trains a recurrent VAE over a large number of randomized tasks in a specific domain (e.g. mazes or car racing tracks) and then performs PAIRED over the learned latent space of the VAE. Their results show that PAIRED in such latent-spaces produces more robust protagonist policies than standard PAIRED, which designs tasks in the direct task space.

**Summary Of The Review:**

This paper provides an interesting extension of UED, in which a PAIRED adversary, which generates regret-maximizing tasks for an RL agent, searches for such tasks in a learned latent space. The idea in itself is interesting, but the current paper lacks important experimental details and makes several ambiguous or unjustified claims. Therefore, I cannot recommend this paper for acceptance.

---

> ### Author Response · Authors · 2022-11-18
> **Response to Reviewer NPYm (4)**
>
>
> **Feedback 5: Curriculum analysis is lacking**
>
> **Curriculum Results:**  We have added a new section in the appendix (Section E.1) with the analysis.
>
>
> **Concern regarding the Number of Blocks for the Minigrid Experiment:** While decoding environments from the decoder, we limit the decoding steps to the maximum number of obstacles, i.e. 50. Hence, none of the generated environments can have more than 50 obstacles.
>
> **Comparison with Domain Randomization(DR):**
>   We added a new Section E.2 in the appendix to show that CLUTR teacher generates distinctively different curriculum than a random policy.
>
>
> **Clarity:**
>
> -   Impact of Flexible/Standard Objective: Please refer to Weakness-Feedback 1-Point 2
>
> -   Number of Seeds for Minigrid Experiments: We ran 5 independent runs of PAIRED and CLUTR. We updated the manuscript with this detail.
>
> -   Confusion about decoder finetuning: The decoder was fine-tuned only for the experiment in section 5.4; in all our other experiments, the decoder was kept fixed during the curriculum learning. We amended section 4.1 last paragraph for better clarity about this point.
>
>
> **Novelty:**
> -   Related work on latent-task space: Thanks for the suggestion. We have updated the related work with the reference.
>
>
> **Reproducibility**
> We have shared our code, training data, and models here: [https://github.com/clutr/clutr](https://github.com/clutr/clutr)

---

> ### Author Response · Authors · 2022-11-18
> **Response to Reviewer NPYm (3)**
>
>
> **Feedback 4: The results do not provide evidence for whether the VAE and CLUTR adversary learn anything**
>
> **SubPoint 1 - Potential Additional Cost of larger action Space:**
>
> While it is difficult to compare discrete and continuous action spaces, depending on the parameterization of the task CLUTR teacher may need to work with a large action space compared to PAIRED. However, CLUTR teachers take only a single step, while the PAIRED teacher requires steps equal to the length of the parameter vector. The difficulty of the underlying RL problem depends both on the size of the action space and the horizon length. Moreover, it is a common consensus that teaching a parameter-space RL teacher is difficult due to long-term credit assignments with sparse rewards [1].
>
> Secondly, we can control the latent dimension and still learn a good reconstruction while we can't change the parameter space without changing the problem, e.g., we cannot change the number of blocks without changing the problem. Hence, the latent space dimension and, thus, the CLUTR action space is controllable; and based on the practicality of the problem, we might control it.
>
>
> Lastly, our experimental results show CLUTR trains better student agents than PAIRED more efficiently, which implies that CLUTR teacher generates a better curriculum than PAIRED. Hence larger action space has not been detrimental.
>
>
>
> [1] Parker-Holder, Jack, et al. "Evolving Curricula with Regret-Based Environment Design." arXiv preprint arXiv:2203.01302 (2022).
>
> **SubPoint 2 - Usefulness of the learned representation:**
>
> We have added two new Sections E.3 and D.4 in the Appendix to address this concern.
>
> Additionally, Representation Learning via VAE has been shown to learn useful representations and help downstream tasks in many domains, including RL, which has been discussed in the ‘Representation Learning’ subsection of the paper. The usefulness of the learned representation can be inferred from the improvement it brought in the downstream task, i.e., CLUTR’s superior performance over PAIRED.
>
>
> **Visualization of the Latent Space Structure:**
>
>
> We added a new section E.3, in the appendix to address this feedback. We linearly interpolated in the latent space between two grids (one empty and the other one is a 15x15 version of the FourRoom grids from the test tasks) and have visualized the reconstruction of the interpolated grids. We show that, as we interpolate in the latent space, the reconstructed grid incrementally adds more obstacles, and the grids start to look more like the target grid. We also note that the reconstruction is not perfect yet reasonable.
>
> **Is CLUTR teacher different than a random policy:**
>
> To address this question, we generated a random-latent curriculum (DR) that uses the same decoder as CLUTR but samples latent vectors randomly rather than using a learned policy (Section E.2 of the appendix). Figure 19 clearly shows that the curriculum generated by the CLUTR teacher is distinctively different from the uniformly random DR curriculum.

---

> ### Author Response · Authors · 2022-11-18
> **Response to Reviewer NPYm (2)**
>
>
> **Feedback 3: Interpretation of Figure 4 and Figure 5**
>
> **Figure 5 Interpretation:**
>
> We would like to clarify that in Figure 5 and Section 5.3, we focus on the efficiency of the curriculum learning algorithm, not the teacher. We agree with the reviewer that if the REGRET is high, such as the maximum possible, the teacher can get the highest reward by generating the same environment till the protagonist starts learning. Hence, the objective for the teacher is easier. But, if the REGRET is too high, the protagonist might need more iterations to catch up with the antagonist, hence hindering the efficiency of the curriculum learning algorithm. We explained this later point in more detail in Section 5.3, and we edited the title of the same section to be more clear.
>
> **Figure 4 Interpretation:**
>
> The solved rate shown in Figure 4 is on Unseen Test levels, not teacher-generated training levels. The main essence of Figure 4 is to give the readers an idea of the sample efficiency of CLUTR compared to PAIRED. For this, we periodically evaluated the agents on a set of test environments and plot the performance. These evaluations were not used for training/validation purposes.

---

> ### Author Response · Authors · 2022-11-18
> **Response to Reviewer NPYm (1)**
>
> We thank the reviewer for this detailed feedback and pointers. We updated our paper to reflect these. We also appreciate their support regarding our motivation, connection to related works and how CLUTR fits in the UED literature, the novelty of using task representation learning in the UED context, and the CarRacing experiments.
>
> **Weakness:**
>
> **Feedback 1: The regret objective used by CLUTR is unclear**
>
>
> **Confusion about regret estimation function:** CLUTR uses the standard PAIRED objective for Minigrid. For CarRacing, we have experimented with both the standard and Flexible PAIRED objectives. The main text presents the results with the Flexible PAIRED objective, and the results with the Standard PAIRED objective are discussed in Appendix D.2.
>
> We have specified the regret objectives in our updated manuscript. (Section 5.1, 1st paragraph, line ~4-6, Section 5.2 1st paragraph.)
>
> **Confusion about CLUTR’s gain over PAIRED:** We show in Appendix D.2 that, similar to the Flexible objective, CLUTR with the standard PAIRED objective also outperforms PAIRED. This shows that CLUTR’s gain over PAIRED is not subject to the choice of regret loss: flexible/standard.
>
> **About Flexible Regret and Robustness Guarantee:** We agree with the reviewer. When the flexible objective is used, CLUTR (and PAIRED) might not hold the robustness guarantee. We have amended the paper (Section 4.3) and added a new section B.2 in the appendix to clarify our robustness guarantees and the practical deviations.
> We also want to mention that we experimented with standard regret too and found better performance than PAIRED, as discussed in Appendix D.2.
>
> **Citation for Flexible Objective:**
>
> Thanks to the reviewer for pointing this out. We updated the reference in our revised manuscript.
>
> **Feedback 2: CLUTR does not have access to the full task space:**
> We agree with the reviewer that, in practice, CLUTR VAE might not have access to the full task space due to practical limitations on training, e.g., the training dataset not having all possible tasks. We would like to clarify that when the decoder is allowed to be finetuned, CLUTR will have access to the full task space similar to PAIRED. The main difference in this context is while PAIRED starts from randomly initialized weights, CLUTR starts from the pretrained weights. In experiments, we found that fixing the decoder performs better than finetuning it, and hence we kept it fixed for our main experiments. We have amended section 4.3 and added a new section B.2 in the appendix to clarify this.

---

> ### Comment · Reviewer_NPYm · 2022-11-18
> **Thank you for the follow-up responses**
>
> I thank the authors for the detailed responses addressing several of my concerns. Thank you for clarifying several points, however some key concerns still remain.
>
> Regarding the regret objective used by CLUTR and comparison to DR:
> - It is concerning that the main experiments for MiniGrid and CarRacing use different objectives for the adversary, and therefore this is a confounder in the way the results are presented. I understand that you included standard PAIRED + latent encoding results in the Appendix, but it seems like the final test performance on the four test tracks is similar to the simple domain randomization baseline in Fig 11. This reinforces the possibility that the CLUTR adversary is not learning any meaningful curriculum, and that its behavior, and thus performance, is similar to domain randomization. The gains on CarRacing seem due to using the flexible PAIRED objective, rather than the standard PAIRED objective. This is a major issue in my view, as it shows the main idea presented in this paper is *not* the driver of the gains presented in the results section.
>
> - A related concern: The current experiments section does not compare to the important baseline of domain randomization on the MiniGrid domain. In the "Replay-Guided Adversarial Design" paper (Jiang et al, 2021), it seems that the DR baseline on mazes with up to 50 blocks, used in your experiments, does better than both PAIRED and REPAIRED.
>
> - Lastly, Figure 18 in the new Appendix makes it appear as if CLUTR is not actually changing the distribution over block counts in any significant way over the course of training. Since CLUTR outputs mazes based on an RL policy, the proper comparison would not be to DR over the latent space, as shown in Figure 20 (b), but to the outputs of randomly initialized policies of the same architecture.
>
> Regarding Fig 5 interpretation:
> - I'm not convinced by this explanation: In the Maze environment, the agent can achieve higher regret on mazes where the goal is closer due to the discounting. Thus, in contrary to the exploration argument you put forth, higher regret would be achieved by placing goals closer to the agent, for which the agent cannot reach. In this sense, high regret levels in the maze domain are easier exploration problems, since the goal would be closer to the agent.
>
> Given these remaining issues, I will keep my current rating for the paper.

---

> > ### Author Response · Authors · 2022-11-19
> > **Response to Followup Feedback (1)**
> >
> > Thanks to the reviewer for their response.
> >
> > **CLUTR and its context in the family of UED algorithms**
> > CLUTR  proposes the inclusion of latent task space in curriculum generation for adaptive teacher curriculum learning algorithms. As an instance of an adaptive teacher curriculum learning algorithm, we used PAIRED. Hence, PAIRED is the primary method we compare in our paper: to show the impact of latent space curriculum design. Our empirical results and analysis evidently show CLUTR is performing better than PAIRED. As for using two different regret-approximation for CarRacing experiments, CLUTR outperforms PAIRED in both. This  suggests the inclusion of latent space curriculum design adds significant benefit for adaptive-teacher curriculum learning algorithms (like PAIRED).
> >
> > DR, on the other hand, uses a random teacher. Other better-performing UEDs, such as Robust PLR[2] or ACCEL[3] also use random generators. We want to clarify that adaptive-teacher/generator approach like PAIRED and CLUTR is fundamentally different from these random-generator-based approaches. and we further want to clarify that *our paper focuses on improving adaptive teachers (e.g. PAIRED) via latent task space. Our empirical results and analysis clearly show CLUTR’s advantage over PAIRED, supporting the usefulness of our idea.*
> >
> >
> > **Gains in CarRacing**:
> > We want to clarify that, CLUTR outperformed PAIRED irrespective of the regret approximation, indicating the performance increase is due to the inclusion of latent space and not the choice of the regret approximation (e.g, standard or flexible).  We would also add Flexible PAIRED results on the CarRacing domain for clarity.
> >
> > **Utility of CLUTR Curriculum and comparison with PAIRED Curriculum (Figure 18)**
> >
> > Thank you for this valuable discussion. We would like to point out several differences in patterns that CLUTR and PAIRED exhibit, which we believe have interesting implications regarding what might be good curriculum learning.
> >
> > In PAIRED, the number of obstacles starts high, drops quickly and stays similar for a significant number of steps, and quickly increases with a narrow interval but no visible peak. On the other hand, in CLUTR, the number of obstacles starts flat, centers around a peak around the middle but still with a wide interval for some number of steps, and the peak drops slightly while the interval stays almost the same. This illustrates that we can achieve a more efficient curriculum learning without making the problem too easy early or without focusing on a narrow interval with a flat distribution later. Instead, we can start with a wide interval and gradually focus on a peak around the middle without making the interval very narrow.
> >
> > **CLUTR learns a distinctively different Curriculum from DR:**
> >
> > We would like to point out that in Figure 20, the number of obstacles for CLUTR is clearly not uniform, and there is a clear mode—distinctively different from a DR-generated curriculum.

---

> > > ### Author Response · Authors · 2022-11-19
> > > **Response to Followup Feedback (2)**
> > >
> > > **Regarding Fig 5 interpretation**
> > > While the adversary's goal is to maximize the regret, maximal regret doesn't necessarily lead to a better curriculum. Consider the case where the regret is maximal due to the antagonist agent optimally solving an environment while the protagonist agent is completely failing. In this case, the adversary is not incentivized to create more challenging or easier environments. When the protagonist agent starts learning, the regret drops and the adversary is incentivized to create more challenging environments. This leads to an automated curriculum. This is also in line with the discussion at the end of Page 2 in the PAIRED paper [1]. While we agree that return depends on the discount factor ($gamma$), number of steps, and maximum number of steps allowed in MiniGrid, it is difficult to claim that placing goal next to start location would always give the maximum regret as this also depends on agent policies which constantly evolve and not necessarily optimal.
> > > Our main point was that while what might be the optimal regret distribution is not clear, we can get some idea by comparing CLUTR and PAIRED results. Considering that CLUTR learns more efficiently than PAIRED (as we showed in Figure-2a, 2b, 3, & 4), regret curves in Figure-5 could point that having lower regret could actually give more efficient learning. We believe this could be due to protagonist agent being relatively stronger, i.e., achieving closer to antagonist agent in performance and hence forcing the adversary to generate more challenging environments more often.
> > >
> > > [1] Michael Dennis, Natasha Jaques, Eugene Vinitsky, Alexandre Bayen, Stuart Russell, Andrew Critch, and Sergey Levine. Emergent complexity and zero-shot transfer via unsupervised environment design. Advances in neural information processing systems, 33:13049–13061, 2020
> > >
> > > [2] Jiang, Minqi, et al. "Replay-guided adversarial environment design." Advances in Neural Information Processing Systems 34 (2021): 1884-1897.
> > >
> > > [3] Parker-Holder, Jack, et al. "Evolving Curricula with Regret-Based Environment Design." arXiv preprint arXiv:2203.01302 (2022).

---

> > > > ### Comment · Reviewer_NPYm · 2022-11-19
> > > > **Response to authors**
> > > >
> > > > ### Regarding comparisons to PAIRED on MiniGrid with 50 blocks
> > > > I disagree here that simply improving PAIRED on the MiniGrid setting with 50 blocks is sufficient to show CLUTR learns a meaningful curriculum. The reason here is that in the paper "Replay-Guided Adversarial Design" (Jiang et al, 2021), which the authors cite, domain randomization actually matches or exceeds PAIRED (Figure 11 in that paper). This previous result suggests this setting of PAIRED simply does not work well on the 50 block domain (one issue is that the original PAIRED results were reported against a fixed domain randomization baseline with 25 blocks, but Jiang et al, 2021 does show PAIRED with 25 instead of 50 blocks outperforms DR on MiniGrid with 25 blocks, just not with 50 blocks). Since your setting is 50 blocks, it would require showing CLUTR outperforms domain randomization.
> > > >
> > > > ### Regarding whether CLUTR learns a meaningful curriculum
> > > > Just showing that CLUTR's weighing over block counts over time follows a different shape to that of PAIRED is not sufficient to show it learns a meaningful curriculum. The important thing to show is that CLUTR learns a curriculum that is doing something different from some random adversary policy. While you compare to domain randomization, this is not a fair comparison: The reason is that the null hypothesis here is that CLUTR's adversarial policy network is not learning meaningful behavior that departs from random. Therefore, to provide evidence against this hypothesis, you must compare CLUTR's adversarial policy against the same policy with random weights or perhaps with random regret estimates, that is, with random returns per environment design episode for the adversary, and show that CLUTR's curriculum leads to a significant improvement to the designs output by such a random network.
> > > >
> > > > ### Regarding the usefulness of Figure 5
> > > > I think this discussion mostly shows that Figure 5 does not convey any information about the effectiveness of CLUTR over PAIRED. If you view minimax regret UED as a zero-sum game, a good way to assess whether the CLUTR student is more robust than the PAIRED student would be to compare their relative performances against *held-out* adversaries. This is precisely what evaluating on zero-shot held-out environments effectively does, and as your results show, it is unclear if the gains from CLUTR on CarRacing are due to using the flexible PAIRED objective, and it is not clear if CLUTR actually learns a meaningful curriculum in MiniGrid, because PAIRED is such a weak baseline in MiniGrid with 50 blocks, where it underperforms domain randomization.
> > > >
> > > > For these reasons, I do not find the current results in this paper to provide sufficient evidence to support the claims made by the authors. Therefore I will keep my current rating of the paper.

---

> > > > > ### Author Response · Authors · 2022-12-10
> > > > > **Follow Up on CLUTR teacher's learning**
> > > > >
> > > > > We wanted to follow up on the main issue raised by the reviewer---whether CLUTR teacher learns anything.
> > > > > We conducted new additional experiments to further provide evidence that CLUTR teacher is indeed learning and different from a random policy.
> > > > >
> > > > > 1) We conducted a CarRacing experiment, where the teacher wasn't updated during the student training, i.e., the initial random teacher generated the environments throughout.  For this experiment we used the same teacher architecture,  as well as the same pretrained VAE, as all other CLUTR experiments. This version of CLUTR with random teacher achieved a mean reward of -38.45 on the Full F1 benchmark, way less than CLUTR which achieves 339 with standard regret and 468 with flexible regret, implying CLUTR teacher trains significantly different agents  than a random teacher. A figure showing the above comparison can be found here:
> > > > > https://github.com/clutr/clutr/blob/main/figs/clutrfp_vs_shuffled_vs_finetuned_no_train_test.png
> > > > >
> > > > >
> > > > > 2) We generated a curriculum by sampling the pretrained latent space with a random CLUTR teacher. The random teacher uses the same architecture and initialization procedure as the original CLUTR teacher, as well as the same pretrained VAE. We found that the random teacher curriculum is significantly different from CLUTR and is similar to the Domain Randomization curriculum on latent space. The figure can be found here: https://github.com/clutr/clutr/blob/main/figs/paper_snap_curriculum.png
> > > > >
> > > > >
> > > > > The above experiments provide further evidence that CLUTR teacher indeed learns and is different from a random teacher.
> > > > >
> > > > >
> > > > > 3) For CarRacing, we also conducted another experiment, where we start with a random VAE and finetune it using the regret loss while simultaneously training the CLUTR teacher.  This version  achieved a mean reward of -17.16 on the full F1 benchmark, way less then CLUTR. This further signifies the the importance of pretraining. A figure showing the above comparison can be found here:
> > > > > https://github.com/clutr/clutr/blob/main/figs/clutrfp_vs_shuffled_vs_finetuned_no_train_test.png

---

> > > > > > ### Comment · Reviewer_NPYm · 2022-12-10
> > > > > > **Follow-up question**
> > > > > >
> > > > > > I have a few follow-up questions to help me understand these results:
> > > > > >
> > > > > > 1. How is the 3D histogram generated for the random teacher? Do you sample from the untrained teacher network to produce each sample?
> > > > > > 2. How does domain randomisation's performance compare against CLUTR/PAIRED on the minigrid environment with 50 blocks?
> > > > > > 3. If the random teacher's 3D histograms look like domain randomisation, why is its performance different from DR on CarRacing?
> > > > > >
> > > > > > Also, it seems it is still unclear if the gain on CarRacing is due to Flexible PAIRED. The performance of around 339 seems to match DR, which makes me think CLUTR is simply randomizing without much of a curriculum, thereby matching the naive domain randomisation baseline.

---

### Author Response · Authors · 2022-11-18
**Common Response**


We thank all the reviewers for their constructive feedback and suggestions. We appreciate the reviewers' support regarding our motivation, connection to related works, the novelty of using latent task representation learning in the UED context, writing, empirical results, and comprehensive experiments. Based on the reviewers’ feedback, we have attempted to address all the raised concerns to the best of our ability, further improving our paper.


Aside from the individual responses to each reviewer, here we would like to comment on the major high-level concerns about the paper:



**Novelty/Contribution:**



Training a teacher with RL is one of the most critical challenges of parameter-space adaptive-teacher UED algorithms, i.e., PAIRED, REPAIRED [1], and often regarded as the Achilles’ heel of adaptive-teacher UEDs [2]. CLUTR makes an important step towards solving this problem by introducing latent space teachers—a novel contribution in this domain. We conducted extensive experiments to evaluate the strengths and drawbacks of our approach. Our experimental results and analysis strongly support the advantage of our approach.


Furthermore, Our approach draws motivation from the recent successful trend of using pretrained networks and improving downstream tasks in domains such as NLP. Our use of a pretrained network and VAE is also novel in its context.



[1] Jiang, Minqi, et al. "Replay-guided adversarial environment design." Advances in Neural Information Processing Systems 34 (2021): 1884-1897.


[2] Parker-Holder, Jack, et al. "Evolving Curricula with Regret-Based Environment Design." arXiv preprint arXiv:2203.01302 (2022).



**Clarity & Correctness:**

A number of concerns raised by the reviewers were due to clarity issues. We have made appropriate changes throughout our paper to address these concerns including rewrites, new discussions, and new empirical analysis to support our paper.


We have also shared our code, training data, and models here: [https://github.com/clutr/clutr](https://github.com/clutr/clutr)

---

### Decision · Program_Chairs · 2023-01-20

**Decision:**

Reject

**Justification For Why Not Higher Score:**

The paper has no strong reasons to be accepted given the concerns about the validity of the proposed improvements. Authors definitely need to have convincing arguments why their method is better than the simple domain randomization baseline.

**Justification For Why Not Lower Score:**

N/A

**Metareview: Summary, Strengths And Weaknesses:**

This paper proposes CLUTR, a curriculum-learning algorithm based on PAIRED which decouples task representation and curriculum learning. The experimental results show that CLUTR performs better that other existing methods.

One of the main criticisms of this paper is about the curriculum learned by the CLUTR. Reviewer NPYm argues that CLUTR is only matching the domain randomization baseline and it needs more analysis to understand what is going on. Given that this paper still requires some work, I recommend rejection and encourage the authors to incorporate the reviewers' comments and suggestions for their future submission.